# BRCA Mutations in Ovarian and Prostate Cancer: Bench to Bedside

**DOI:** 10.3390/cancers14163888

**Published:** 2022-08-11

**Authors:** Stergios Boussios, Elie Rassy, Michele Moschetta, Aruni Ghose, Sola Adeleke, Elisabet Sanchez, Matin Sheriff, Cyrus Chargari, Nicholas Pavlidis

**Affiliations:** 1Department of Medical Oncology, Medway NHS Foundation Trust, Windmill Road, Gillingham ME7 5NY, UK; 2Faculty of Life Sciences & Medicine, School of Cancer & Pharmaceutical Sciences, King’s College London, London SE1 9RT, UK; 3AELIA Organization, 9th Km Thessaloniki-Thermi, 57001 Thessaloniki, Greece; 4Department of Medical Oncology, Gustave Roussy Institut, 94805 Villejuif, France; 5Novartis Institutes for BioMedical Research, CH 4033 Basel, Switzerland; 6Department of Medical Oncology, Barts Cancer Centre, St. Bartholomew’s Hospital, Barts Health NHS Trust, London E1 1BB, UK; 7Department of Medical Oncology, Mount Vernon Cancer Centre, East and North Hertfordshire NHS Trust, London KT1 2EE, UK; 8Centre for Education, Faculty of Life Sciences and Medicine, King’s College London, London SE1 9RT, UK; 9High Dimensional Neurology Group, UCL Queen’s Square Institute of Neurology, London WC1N 3BG, UK; 10Department of Oncology, Guy’s and St Thomas’ Hospital, London SE1 9RT, UK; 11School of Cancer & Pharmaceutical Sciences, King’s College London, Strand, London WC2R 2LS, UK; 12Department of Urology, Medway NHS Foundation Trust, Windmill Road, Gillingham ME7 5NY, UK; 13Medical School, University of Ioannina, Stavros Niarchou Avenue, 45110 Ioannina, Greece

**Keywords:** DNA damage repair, homologous recombination, PARP inhibitors, ovarian cancer, prostate cancer

## Abstract

**Simple Summary:**

DNA damage is one of the hallmarks of cancer. Epithelial ovarian cancer (EOC) —especially the high-grade serous subtype—harbors a defect in at least one DNA damage response (DDR) pathway. Defective DDR results from a variety of lesions affecting homologous recombination (HR) and nonhomologous end joining (NHEJ) for double strand breaks, base excision repair (BER), and nucleotide excision repair (NER) for single strand breaks and mismatch repair (MMR). Apart from the EOC, mutations in the DDR genes, such as *BRCA1* and *BRCA2*, are common in prostate cancer as well. Among them, *BRCA2* lesions are found in 12% of metastatic castration-resistant prostate cancers, but very rarely in primary prostate cancer. Better understanding of the DDR pathways is essential in order to optimize the therapeutic choices, and has led to the design of biomarker-driven clinical trials. Poly(ADP-ribose) polymerase (PARP) inhibitors are now a standard therapy for EOC patients, and more recently have been approved for the metastatic castration-resistant prostate cancer with alterations in DDR genes. They are particularly effective in tumours with HR deficiency.

**Abstract:**

DNA damage repair (DDR) defects are common in different cancer types, and these alterations can be exploited therapeutically. Epithelial ovarian cancer (EOC) is among the tumours with the highest percentage of hereditary cases. *BRCA1* and *BRCA2* predisposing pathogenic variants (PVs) were the first to be associated with EOC, whereas additional genes comprising the homologous recombination (HR) pathway have been discovered with DNA sequencing technologies. The incidence of DDR alterations among patients with metastatic prostate cancer is much higher compared to those with localized disease. Genetic testing is playing an increasingly important role in the treatment of patients with ovarian and prostate cancer. The development of poly (ADP-ribose) polymerase (PARP) inhibitors offers a therapeutic strategy for patients with EOC. One of the mechanisms of PARP inhibitors exploits the concept of synthetic lethality. Tumours with *BRCA1* or *BRCA2* mutations are highly sensitive to PARP inhibitors. Moreover, the synthetic lethal interaction may be exploited beyond germline *BRCA* mutations in the context of HR deficiency, and this is an area of ongoing research. PARP inhibitors are in advanced stages of development as a treatment for metastatic castration-resistant prostate cancer. However, there is a major concern regarding the need to identify reliable biomarkers predictive of treatment response. In this review, we explore the mechanisms of DDR, the potential for genomic analysis of ovarian and prostate cancer, and therapeutics of PARP inhibitors, along with predictive biomarkers.

## 1. Introduction

Spontaneous DNA damage occurs on the order of 10^4^–10^5^ events per cell per day, and it is considered to have a causal role in aging. This includes spontaneous/endogenous genotoxic stress, as well as environmental/iatrogenic sources of genotoxic stress [1]. Endogenous sources of DNA damage and chromatin organization contribute to mutational processes that have been recorded in cancer genomes. Moreover, metabolism is a crucial cellular process that can become harmful for cells by leading to DNA damage. This can occur by an increase in oxidative stress or through the generation of toxic byproducts. In contrast, sources for exogenous DNA damage are rare and include ionizing and ultraviolet radiation, as well as various chemicals agents. Different mutational processes generate unique combinations of mutation types, termed “mutational signatures”. In the past few years, large-scale analyses have revealed many mutational signatures across the spectrum of human cancer types [2,3]. Genomic instability can arise from a genetic or epigenetic mutation in a mutator gene such as in a DNA damage repair (DDR) gene [4]. Several mechanisms can be activated to repair damaged DNA, including homologous recombination (HR) repair, nonhomologous end joining (NHEJ), base excision repair (BER), nucleotide excision repair (NER), and mismatch repair (MMR) [5,6]. HR is the main mechanism for high-fidelity repair of double-strand DNA breaks (DSB) [7]. Mutations in genes related to this pathway may lead to HR deficiency. Among them, *BRCA1/2* mutations are the most frequent and lead to hereditary breast and epithelial ovarian cancer (EOC). Hereditary breast and ovarian cancer due to mutations in these genes is the most common cause of hereditary forms of both breast and ovarian cancer, accounting for 30–70% and approximately 90% of cases, respectively [8]. In individuals harboring mutations in *BRCA1/2* genes, the probability of developing breast cancer over a lifetime is around 85%, and that of EOC is about 20–40% [9]. *BRCA1* and *BRCA2* mutation carriers are mostly single heterozygous with only one mono-allelic deleterious mutation on one of these two genes. Excluding individuals of Ashkenazi descent, it is uncommon to identify carriers of two deleterious mutations either within the same gene (biallelic) or in both genes (trans-heterozygous). Trans-heterozygous mutations in both *BRCA1* and *BRCA2* genes are clinically correlated with an early age of onset and a severe disease compared to single heterozygous *BRCA* mutation carriers. Breast and ovarian cancer risks differ depending on the position and the type of *BRCA1* and *BRCA2* mutations. Importantly, two different mutations on the same allele may be associated with a distinctive phenotype, since each mutation is located in a different domain of the BRCA protein. Consequently, the interaction of BRCA with several other proteins could be disturbed. Therefore, these altered protein-protein interactions may impact on the phenotype. The *BRCA* mutation location also affects the EOC risk. *BRCA1* and *BRCA2* have been identified in the ovarian cancer cluster region in or near exon 11, and in the breast cancer cluster region in multiple regions other than exon 11 so far. In a recently published report, the authors presented the distribution of the age at diagnosis of EOC with *BRCA* mutation in detail, and analyzed the age by each common mutation type in a Japanese population [10]. The most common mutation in *BRCA1* was *L63X*, followed by *Q934 X*, *STOP799*, and *Y1853C*. Among them, *L63X* and *Y1853C* were located in the breast cancer cluster region, whereas *Q934 X* and *STOP799* were in the ovarian cancer cluster region. As far as the *BRCA2* mutations are concerned, the most common was *R2318X*, followed by *STOP1861*, *Q3026X*, *S1882X*, *P3039P*, *STOP613*, *S2835X*, and *STOP2868*. Among them, *R2318X*, *STOP1861*, and *S1882X* were located in the ovarian cancer cluster region, whilst *S2835X* and *STOP2868* were located in the breast cancer cluster region. Finally, *Q3026X*, *P3039P*, and *STOP613* were not located in either the ovarian or breast cancer cluster regions. Moreover, the majority of serous papillary peritoneal carcinoma are high-grade tumours, and thus present *p53* and *BRCA* mutations [11]. A number of additional variants in genes beyond *BRCA1/2* have been identified and are suspected to play a significant role in ovarian carcinogenesis. Approximately 20% of castration-resistant prostate cancer patients harbour germline or somatic mutations in one of the DDR genes, which supports the mechanism of synthetic lethality [12]. The two main composite HR deficiency tests available in clinical practice apply next-generation sequencing (NGS) or microarray assays to simultaneously search for *BRCA* mutations and genomic scars.

From the therapeutic point of view, targeting the DDR pathway is a reasonable approach. Within this context, several poly (ADP-ribose) polymerase (PARP) inhibitors have been considered for the treatment of several malignancies, including EOC and prostate cancer. Based on the successful application of PARP inhibitors in BRCA-deficient breast cancer and EOC, PARP inhibitors are currently being investigated for the treatment of metastatic prostate cancer with promising results [13]. In this review, we discuss the current landscape of genetic testing and management of the hereditary risk for EOC and prostate cancer, and the application of PARP inhibitors in the precision treatment of these clinical entities. Furthermore, we highlight the importance of developing predictive biomarkers for the optimal selection of the patients who benefit from the PARP inhibitors.

## 2. Molecular Landscape

DNA damage is a frequent event during cell life and can be spontaneous or caused by cell metabolism or by environmental agents. There have been six primary pathways of DNA repair identified, which are variably used to address DSB and single-stranded DNA break (SSB) damage from a variety of mechanisms of injury. HR and NHEJ recombination are the two major pathways responsible for repairing DSB, whereas the primary mechanisms for resolving SSB are the BER, NER, MMR, and translesional synthesis [5,8]. The function of the primary DDR pathways begins with sensing DNA damage. The next step is the recruitment of proteins involved in building the repair complexes [12]. The potential absence, reduction, or dysfunction of these proteins may result in loss of function of proper DDR. HR pathways become active in the S/G2 phase due to the availability of a sister chromatid, whereas NHEJ repairs DSB throughout all cell cycle phases except the M phase. DSB end resection directs the pathway towards HR during the S/G2 phase. Apparently, only 30% of DSB undergo resection, and hence HR in the G2 phase. On the other hand, during the S phase, DSB are mainly repaired by HR, although two-ended DSB may still be repaired by NHEJ, unless the replication machinery encounters the DSB ends. In addition to cell cycle-dependent regulation, DSB end complexity is critical for directing preferential repair by HR. SSB normally do not compromise the integrity of DSB. However, if an SSB is left unrepaired and the lesion is encountered by DNA machinery that separates the DNA duplex into two component SSB, an SSB can be converted into a one-ended DSB [14]. SSB and DSB also arise during aberrant DNA topoisomerase reactions, spontaneously or upon exposure to specific inhibitors [15]. Break-induced replication (BIR) is one of the pathways that drives genome instability, as it results in a loss of heterozygosity, mutations, and nonreciprocal translocations [16]. In fact, DSB at collapsed forks are single ended, with no second end available for classical HR repair. These breaks can be processed by BIR, a conservative DNA synthesis mechanism described as an HR-based repair pathway for one-ended DNA DSB [17].

### 2.1. Homologous Recombination and Nonhomologous End Joining

DSB are one of the most common and cytotoxic types of DDR associated with significant genomic aberrations, which if left unrepaired or improperly repaired may lead to cell death. DSB are repaired by a number of repair pathways, the most important of which involve HR and NHEJ [8,13].

HR is restricted to the S and G2 phases of the cell cycle due to the cell cycle-dependent availability of sister chromatids. This is also correlated with the fact that cyclin-dependent kinases (CDKs) have a modulatory role on DSB components, including their influence on enzymes involved in HR [18]. NHEJ is an error-prone process that simply fuses the two broken ends together, whereas HR is error-free, using the genetically identical sister chromatid as a template for repair. This statement is largely based on the fact that the mechanism of HR requires the search for a homologous partner to repair DNA, in contrast to NHEJ. However, nowadays, that position has been reconsidered. The products of HR are gene conversion, associated or not with crossing-over. Such products can account for genetic diversity or instability arising through HR. Gene conversion may transfer genetic information in a non-reciprocal manner between two hetero-alleles, resulting in a loss of heterozygosity. It can also transfer one stop codon from a pseudogene to a related coding sequence, leading to its extinction. Mus81 and Yen1 endonucleases, as well as Slx4, promote replication template switching during BIR, thus participate in the generation of complex rearrangements when repeated sequences dispersed throughout the genome are involved [19]. NHEJ is faster than HR and mainly occurs in the G1 phase [20]. Nevertheless, there is recent evidence that NHEJ functions throughout the cell cycle. Beyond the already-known proteins, such as Ku70/80, DNA-PKcs, Artemis, DNA pol λ/μ, DNA ligase IV-XRCC4, and XLF, new proteins are involved in the NHEJ, namely PAXX, MRI/CYREN, TARDBP of TDP-43, IFFO1, ERCC6L2, and RNase H2. Among them, MRI/CYREN has dual role, as it stimulates NHEJ in the G1 phase of the cell cycle, while it inhibits the pathway in the S and G2 phases [21]. The extent of DNA end resection is the primary factor that determines whether repair is carried out via NHEJ or HR. 53BP1 and the cofactors PTIP or RIF1-shieldin protect the broken DNA end, inhibit long-range end resection, and thus promote NHEJ. The cell cycle, the chromatin environment, and the complexity of the DNA end break affect the DNA resection [22]. In HR repair, the nuclease meiotic recombination 11-like (MRE11) forms a complex with RAD50 and NBS1 (Nijmegen breakage syndrome 1)—MRN complex—which detects double-strand breaks (DSBs) and recruits and activates ATM at DNA ends [23,24]. Replication protein A (RPA) is the major protein that binds ssDNA with a high affinity. ATR interacts with a partner ATRIP and recognizes RPA-covered ssDNA (RPA-ssDNA). For the activation of the ATR, apart from the recruitment of the ATR-ATRIP complex (ATR-ATRIP) to RPA-ssDNA, checkpoint regulators—such as TopBP1 and ETAA1—participate as well. TopBP1 is recruited to sites of DNA damage or stalled replication forks and engage with the Rad9-Rad1-Hus1 complex at dsDNA-ssDNA junctions, leading to stimulation of the ATR-ATRIP kinase. Similarly, ETAA1 directly activates ATR-ATRIP. Thus, ATR-ATRIP is recruited by recognizing RPA-ssDNA and subsequently activated in a multiple-step process. In budding yeast, the Mec1-Ddc2 complex (Mec1-Ddc2) corresponds to ATR-ATRIP [25]. Single-stranded DNA generated by resection is coated by RPA, which recruits Ddc2 and the Mec1 checkpoint kinase [26]. That leads to the formation of nucleoprotein filaments in ssDNA, which are essential for the homology search in sister chromatid and strand exchange [27]. KAT5 is an acetyltransferase, participating—among others—in transcriptional regulation, chromatin remodeling, histone acetylation, and DNA repair. The ability of KAT5 to regulate chromatin structure at DSB is mediated through its interaction with a multifunctional remodeling complex (NuA4 complex), which is recruited to DSB. The NuA4-KAT5 complex acetylates histones H2AX and H4 at DSB, and modifies chromatin architecture to facilitate DSB repair. The phosphorylation of the c-terminal of the histone variant H2AX by ATM is crucial in DSB repair. H2AX is rapidly phosphorylated on chromatin domains surrounding the DSB. Then, the mdc1 scaffold protein binds directly to γH2AX and formulates a platform for the recruitment of other DNA repair proteins, including BRCA1 at the DSB [28].

A number of factors have been believed to stabilize the NHEJ complex at DSBs including Ku, the DNA-PK catalytic subunit (PKcs), and the kinase activity of DNA-PKcs. This multi-unit complex recruits further proteins, such as Artemis and PNK, to repair it into a normal DNA structure. The biochemical activity of DNA ligases results in the sealing of breaks between 5′-phosphate and 3′-hydroxyl termini within a strand of DNA [29].

### 2.2. Synthetic Lethality

In 2005, the idea of synthetic lethality in the DDR pathway, via *BRCA1* mutation, was published by two groups [30,31]. The concept of synthetic lethality applies when two non-lethal defects combine and result in a lethal phenotype. Inactivation of one gene allele by mutation and inhibition is not toxic for cells. The synthetic lethality strategy requires gene pairs that, when simultaneously inactivated, cause cell death. Figure 1 demonstrates the genetic landscape of synthetic lethality. However, if *BRCA1* is mutated, PARP inhibition would prevent DDR and lead to tumour cell apoptosis. Therefore, inactivation of both demonstrates synthetic lethality. PARP inhibition specifically in *BRCA1/2* deficient tumour cells can result in up to a 1000-fold increased sensitivity as compared to BRCA wild-type tumour cells [30]. Synthetic lethality could also be used in tumours which share molecular features of *BRCA* mutated tumours—known as “BRCAness”. Therefore, mutation of genes beyond *BRCA* in the HR pathway expands the indication of PARP inhibitors. The broader use of synthetic lethality targeting the HR pathway is still being investigated [32].

Inactivation of Rad52 in BRCA2-deficient cells results in synthetic lethality, which makes Rad52 a tumour-specific target for therapy in BRCA2-deficient tumours. As Rad52 is required for cellular proliferation in BRCA2-defective cells, silencing of Rad52 could cause BRCA2-defective tumour cells. Therefore, inactivation of Rad52 could be reasonable approach for the treatment of a BRCA-defective subset of tumours. Rad52 also represents a potential therapeutic target because no *Rad52* mutations or inactivation has been documented in tumours. Other synthetically lethal relationships have been reported for Rad52, with X-ray repair complementing defective repair in XRCC3 [33]. Recent studies showed that Rad52 is involved in multiple DDR pathways, including a BRCA-independent HR repair involving Rad51, single-strand annealing (SSA), BIR repair, RNA-templated DSB repair, and transcription-associated HR involving XPG. However, how Rad52 activity helps HR-deficient cancers to survive is unclear. A recently published study demonstrated that inhibition of RPA:RAD52 protein-protein interaction (PPI) appears to inhibit Rad52-mediated DNA repair, and that mitoxantrone may be a potent Rad52 inhibitor [34]. It has also been proposed that targeting of Rad52 with small molecule inhibitors will disrupt the Rad52-dependent HR sub-pathway in BRCA1- and BRCA2-deficient cells, causing their lethality. In a study, the selected inhibitors of two different chemotypes exhibited an inhibitory effect on tested BRCA1- and BRCA2-deficient cells [35]. More recently, synthetic lethal interactors of *BRCA1/2* have been identified, including DNA polymerase theta (*POLQ*), flap structure-specific endonuclease 1 (*FEN1*), and apurinic/apyrimidinic endodeoxyribonuclease 2 (*APE2)* [36]. The function of Rad52 in replication fork repair following stress is still not clear. Genotoxic stress results in cell cycle arrest, which is implemented by checkpoint kinases CHK1 and CHK2. CHK1 mainly responds to short replication stress, whereas CHK2 is activated during chronic stress leading to DSB formation. CHK1 activation occurs via phosphorylation by ATR. Active CHK1 targets Cdc25 A phosphatase, which upon phosphorylation undergoes proteasomal degradation. This leads to the reduction of Cdk2/Cyclin A complex activity, conferring checkpoint arrest. CHK1 protein also interacts with BRCA2 and Rad51 proteins, directly phosphorylating them for the formation of active Rad51 nucleofilament, especially during replication blockage. It has been shown that Rad52 overexpression in BRCA2 deficient cells leads to restoration of checkpoint arrest during replication stress and the mitigation of excess origin firing observed in BRCA2 deficient cells [37]. It is still unknown whether PALB2 participates in recruiting and regulating Rad52 in RAD51-mediated HR. The specific genetic interactions of BRCA1 and PALB2 with Rad52 have not yet been clarified; nevertheless, it is considered that BRCA1 and PALB2 are independent of Rad52, which would be compatible with a synthetically lethal relationship. However, it is also possible that Rad52-Rad51-driven HR could be dependent on BRCA1 or PALB2. Repair of DSB relies upon the BRCA1-PALB2-BRCA2 pathway, with Rad52 functioning in an alternative pathway that mediates Rad51-directed repair when deficiencies exist in BRCA1, PALB2, or BRCA2. Nevertheless, a cooperative role for Rad52 and BRCA2 in mediating RAD51 function cannot be definitively excluded [38].

Conversely, certain *Rad52* mutations rescue *BRCA2* mutations. *hRad52 S346X* is a mutation that codes for a Rad52 protein, with 17.2% of its amino acid sequence absent from its C terminus. In a study, this mutation was found to protect against the development of breast cancer in *BRCA1/2* mutation mutants [39]. *hRad52 S346X* also suppressed the elevated frequency of SSA caused by reducing the level of *BRCA2*. Therefore, *hRad52 S346X* may suppress tumourigenesis in BRCA2-deficient cells by suppressing the mutagenic effects of elevated SSA. Alternatively, the suppression of SSA by *hRad52 S346X* may block tumour formation in BRCA2-deficient cells, given the loss of two mechanisms of DSB repair. The observation that *Rad52 S346X* decreases the risk of cancer in BRCA mutants suggests that Rad52 inhibitors may also be a tool for the reduction of breast cancer risk in this subset of patients. However, the toxicity and impact of long-term use of such inhibitors should be further evaluated [40].

### 2.3. Mismatch Repair (MMR)

Base mismatches and insertion-deletion loops (IDLs) that occur during replication are repaired by the MMR pathway, demonstrated in Figure 2 [41]. MMR reduces DNA errors 100–1000 fold, and prevents them from becoming fixed mutations during cellular proliferation [42,43]. The role of MMR defects in the development of cancer was first established when mutation in *MSH2* was linked to hereditary nonpolyposis colorectal cancer, also known as Lynch syndrome [44]. Over time, the MMR genes *MLH1*, *MSH2*, *MSH6*, and *PMS2* were associated to an autosomal dominant, hereditary predisposition to colon cancer. These cancers were hallmarked by germline loss-of-function alterations. Given that prostate is not a Lynch-associated cancer, a focus on somatic mutations leading to the deficiencies in MMR prostate cancer phenotype is reasonable. There are eight MMR genes that have been investigated so far; *hMSH2*, *hMSH3*, *hMSH5*, *hMSH6*, *hMLH1*, *hPMS1 (hMLH2)*, *hMLH3*, and *hPMS2* (*hMLH4*). The prevalence of deficiencies in MMR in prostate cancer has been reported between 3% and 5%. Among the MMR genes, defects within the *MSH2* and *MSH6* gene have been the most frequently reported in patients with prostate cancer. In contrast to other cancers, complex structural rearrangements appear to be an important cause of deficiencies in MMR in prostate cancer. The product of the gene *hMSH2* is the principal corrective MSH protein. The MSH2/MSH6 and MSH2/MSH3 heterodimers function as sensors, recognizing mismatched DNA [45]. The predominance of the heterodimer MSH2/MSH6 is explained by the fact that MSH6 is expressed 10 times more than MSH3. The heterodimer MSH2/MSH6 initiates the repair of small IDLs, while MSH2/MSH3 repairs larger IDLs, up to 13 nt in size [46]. Patients with deficiencies in MMR are eligible for immune checkpoint inhibitor therapy in second-line treatment for metastatic castration-resistant prostate cancer [47].

Specific mutational signatures have been identified in tumours with mutations in the exonuclease (proofreading) domain of polymerase epsilon (POLE), as well as tumours with mutations or epigenetic silencing of MMR genes. Initially, the position was that simultaneous loss of both POLE or polymerase delta (POLD1) proofreading and MMR function could not be tolerated by cells due to excessive accumulation of mutations. However, an analysis of tumours from children with biallelic germline MMR deficiency demonstrated a subset of tumours with remarkably high mutation burdens (>250 mutations/Mb) that also had a somatic mutation in POLE or POLD1 [48].

Malfunctioning of MMR proteins, due either to mutation or reduced expression, suggests the correlation of cancer development to the aberrations of all or the majority of MMR proteins. In EOC, MMR deficiency is the second common cause of hereditary ovarian cancer—only behind HR deficiency—accounting for 10–15% of hereditary ovarian carcinomas [42]. Apart from the inherited gene mutations, additional mechanisms of gene inactivation leading to loss of expression of one of the main MMR genes occurs in up to 29% of cases [42]. Furthermore, it has been reported that high mRNA levels of MSH6, MLH1, and PMS2 were associated with a prolonged overall survival (OS) in EOC. That supports the potential positive prognostic value of MMR genes in EOC patients treated with platinum-based chemotherapy [49]. Prostate cancer has high prevalence of DDR genes alterations. In the metastatic setting, the Prostate Cancer Foundation-Stand Up To Cancer (SU2C-PCF) team identified a high proportion of actionable mutations, including 23% with mutations and other alterations in DDR genes, such as *BRCA2* (13%), ATM (7.3%), and *BRCA1* (0.3%), along with mutations in MMR genes, such as MSH2 (2%) [50]. Recently, the first phase I dose-escalation study of the ATM inhibitor M3541 in combination with palliative radiotherapy in patients with advanced solid tumours was published [51]. M3541 had pharmacokinetic limitations that prevented an analysis of efficacy. Although the clinical development of M3541 was halted, the ATM pathway still represents an attractive therapeutic target. Indeed, the development of the second-generation ATM inhibitor M4076 is ongoing [52]. As far as the ovarian cancer is concerned, ATARI is the first clinical trial aiming to determine whether a synthetic lethal interaction between the tumour suppressor gene *ARID1A* and *ATR* translates into improved outcomes [53]. The inclusion of the olaparib combined with the ATR inhibitor ceralasertib will investigate whether the two classes of DDR inhibitors have the potential to provide clinical activity. The study trial will also aim to identify novel biomarkers of ATR inhibitor response and resistance.

## 3. Susceptibility to EOC

Around 20–25% of unselected EOC patients carry pathogenic variants (PVs) in a number of genes that mostly encode for proteins involved in DDR pathways [54]. Indeed, NGS revealed that beyond *BRCA1/2*, mutations in HR effectors, such as *PALB2*, *RAD51*, *ATM*, *BRIP1*, *BARD1*, and *C**HEK2* occurs in up to a fifth of the patients with high-grade serous ovarian cancer [55]. PVs in each of these genes is associated with variable risk for EOC development. Moreover, mutated or downregulated *ARID1A* significantly compromises HR repair of DNA DSBs [56]. ARID1A is recruited to DSBs via its interaction with ATR. ATR inhibition in ARID1A defective cells thus increases large scale genomic rearrangements and ultimately causes cell death. Loss of function mutations in *ARID1A* leading to a loss of protein expression are a frequent observation in EOC of clear cell and endometroid histology. Finally, PVs genes involved in the MMR pathway account for 10–15% of hereditary EOC, typically in endometrioid or clear-cell histological subtypes [57].

### 3.1. BRCA1 and BRCA2 Genes in EOC

PVs in *BRCA1/2* genes are detected in 10–15% of unselected EOC patients, accounting for the majority of hereditary cases [58]. *BRCA1* and *BRCA2* PVs confer 44% and 17% lifetime risks for EOC diagnosis, respectively, whilst the relevant risk in the general population is approximately 1% [59]. The carriers of *BRCA1* and *BRCA2* PVs are specifically susceptible to develop high-grade serous ovarian cancer, with a median age of onset at about 51 and 61 years, respectively [60]. PVs in *BRCA1/2* in EOC are correlated with prolonged OS, visceral disease distribution, higher response rates to platinum-based chemotherapy, and sensitivity to PARP inhibitors.

The identification of *BRCA1* and *BRCA2* PVs is recommended as an effort of primary prevention for EOC. Cascade screening is the systematic identification and testing of relatives of a known mutation carrier. This strategy determines whether asymptomatic relatives also carry the known variant, in view of offering risk-reducing options to reduce the morbidity and mortality. On the other hand, all negatively tested family relatives have a lifetime EOC risk, compatible with the general population.

The proposed primary prevention strategy for EOC risk reduction among *BRCA1* and *BRCA2* carriers is bilateral salpingo-oophorectomy (BSO), which can reduce the risk for EOC diagnosis up to 96% [60]. *BRCA1* carriers are considered to undergo BSO after the age of 35 and before the age of 40 years, whereas in those with *BRCA2* mutations, BSO can be proposed after the age of 40 and before the age of 45 years, due to the lower penetrance and later onset of diagnosis [61]. The fallopian tube has been established as the origin of the majority of high-grade serous ovarian cancers. The serous tubal intraepithelial carcinoma (STIC) theory originated based on the observation of the presence of occult lesions on the fallopian tubes of women with *BRCA1/2* mutations following prophylactic surgery [62]. That led to the consideration of salpingectomy with ovarian retention until the age of natural menopause within the context of primary prevention [63].

### 3.2. Beyond BRCA1 and BRCA2 Genes in Ovarian Cancer

Besides RAD51 and its meiotic counterpart DMC1, five additional mammalian paralogs of bacterial RecA were discovered two decades ago. RAD51B, RAD51C, and RAD51D were discovered based on DNA sequence alignments, and XRCC2 and XRCC3 through functional complementation of the ionizing radiation sensitivity of Chinese hamster mutant cells. These proteins display limited sequence homology to each other and to RAD51, and are generally reported as classical RAD51 paralogs. Classical RAD51 paralogs were proposed to form two biochemically and functionally distinct subcomplexes, i.e., the RAD51B-RAD51C-RAD51D-XRCC2 complex (BCDX2) and the RAD51C-XRCC3 complex (CX3), showing common, but also distinct, biochemical properties (Figure 3). All human RAD51 paralogs were shown to associate with nascent DNA, but the mechanistic roles of these factors in replication were not investigated systematically. Chinese hamster ovary or DT40 cell lines carrying different mutations in individual genes displayed specific defects in replication fork progression and stability. The paralogs play roles at the early and late stages of HR. The inability of cell lines deficient in RAD51C, XRCC2, and XRCC3 to form damage-induced RAD51 nuclear foci suggests that these three proteins are important in the homology search and strand invasion phase of HR. Except for RAD51B, RAD51 paralog mutants have been isolated and characterized in Chinese hamster cells. In human cells, however, only XRCC3 ablated cells generated in the human colon carcinoma HCT116 cell line have been reported to date. RAD51 paralogs have been implicated in the prevention of aberrant mitoses and aneuploidy, RAD51B, RAD51C, and XRCC3 are implicated in cell cycle checkpoint, RAD51D and XRCC3 in telomere maintenance, and RAD51C, XRCC2, and XRCC3 in termination of gene conversion tracts.

While patients with biallelic mutations in RAD51 and its mediators have been identified in patients with Fanconi anaemia, monoallelic germline mutations in RAD51 mediators are correlated to predisposition to cancer. This is thought to be frequently caused by a somatic loss of heterozygosity (LOH) event, where the second functional copy of the gene is deleted, resulting in genomic instability. Both *RAD51C* and *RAD51D* are EOC susceptibility genes; nevertheless, hereditary ovarian cancer is most commonly caused by a mutation in *BRCA1/2* genes. The prevalence of *RAD51C* loss-of-function germline PVs varies between 0.3% and 1.1%. The lifetime risk of EOC among *RAD51C* carriers is approximately 5% [64]. Damaging *RAD51D* variants are marginally less frequent than *RAD51C* PVs among EOC patients, observed in approximately 0.2% of unselected cases [65]. The relevant percentage in those with strong family history is up to 0.9% [66]. According to the National Comprehensive Cancer Network (NCCN) guidelines, BSO may be considered in the scenario of PVs in any of these genes [67]. Furthermore, genetic defects in *RAD51C* and *RAD51D* genes can function as biomarkers for PARP sensitivity.

Several germline PVs in the so-called moderate- and low-penetrance genes have been associated with a moderate lifetime risk of EOC. The prevalence of *BRIP1* gPVs among familial EOC patients has been reported to be 0.7%, whilst the relevant cumulative lifetime risk among *BRIP* mutants has been estimated as 5–5.8%, predominantly following menopause [68,69]. The elevated risk for EOC diagnosis justifies recommendation for salpingo-oophorectomies among asymptomatic carriers, in line with family history and individual’s preference.

*PALB2* PVs are quite rare among EOC patients, identified in less than 0.5% of cases interrogated [65,68]. Biallelic mutations in PALB2 cause Fanconi anemia subtype FA-N, whereas monoallelic mutations predispose to breast, ovarian, and pancreatic familial cancers [70]. Clinical testing for *PALB2* in EOC is not currently recommended, but can be considered in cases with strong family history for ovarian cancer. The majority of studies reported relative risks between 0.9 and 5.5, but lacked statistical power [67]. It has been observed that *PALB2* associated tumours are sensitive to platinum-based chemotherapy and PARP inhibitors [71].

Furthermore, about 1% of all EOC cases can be attributed to PVs in MMR genes [72]. *MSH2* PVs seem to be associated with the higher risk of 10–24%, followed by *MLH1* PVs with a relevant risk of 5–20%, by the age of 70 years [73]. Finally, *MSH6* PVs confer much lower risks, ranging from 1–11%. In contrast, *PMS2* PVs are not associated with increased EOC risk [73]. Finally, in terms of the non-EOC, a subset of germ-cell tumours can acquire *KRAS*-activating mutations and other genetic alterations, such as *BRCA1/2*, *KIT*, and *MAPK*. However, the efficacy of targeted therapy and genomic features contributing to chemoresistance still remain to be elucidated [74]. Similarly, even though the rate of *BRCA1* and *BRCA2* mutations in ovarian carcinosarcomas is difficult to be determined, the genomic sequencing in some studies has demonstrated loss of function mutations in HR genes. As such, PARP inhibition may be effective even in ovarian carcinosarcomas [75].

### 3.3. Tumour Testing in Ovarian Cancer

Germline and somatic genetic alterations can be identified through tumour testing. On the other hand, germline analysis alone cannot detect somatic mutations. However, there are several factors that may impact the accuracy of tumour testing, meaning it cannot be considered as a gold standard for the potential identification of germline variant detection. Firstly, in most cases, the initial material for tumour testing is DNA extracted from formalin-fixed paraffin-embedded tissues, which can be technically challenging to amplify. Tumour microdissection is required to obtain tumour DNA. An additional concern is tumour heterogeneity and the actual percentage of tumour cells that are included in the tumour specimen from which DNA will be extracted.

If a mutation is identified, the sequencing of the normal cells is required to clarify whether the mutation is germline or somatic. The order by which the germline and somatic BRCA mutational analysis should be performed is an ongoing concern. Germline testing remains in the first place. When initial healthy cell sequencing discards germline PVs, tumour testing should be performed, since somatic mutations can influence treatment decisions. However, there are countries where somatic BRCA testing is performed first, as a screening tool. Patients found to have a mutation in the tumour are then referred to the Genetics team for germline genetic analysis. For those without identified mutations in the tumour, further germline BRCA testing is not recommended. The American Society of Clinical Oncology (ASCO) and the European Society for Medical Oncology (ESMO) guidelines suggest germline testing, followed by somatic tumour testing for the non-carriers of a germline PV. Although trials of EOC treatment stratify patients based on HR deficiency assays, ASCO guidelines do not recommend its routine use [76].

Enumeration of circulating tumour cells (CTC) in the blood may stratify the patients into high- and low-risk groups and serve as a prognostic biomarker for OS and progression-free survival (PFS) in several malignancies. Recent studies have also revealed that characterization of CTC could help predict treatment response. Whether CTC detection is associated with prognosis in EOC remains controversial. Within this context, the standardization of CTC detection techniques is of great importance. Many markers have been applied to the enrichment and screening of EOC CTC. The positive rate of CTC in EOC patients was 60% in advanced stage disease in a study using the CellSearch system targeting EpCAM+ [77]. The detection of CTC using immunomagnetic CTC enrichment targeting EpCAM and MUC1, followed by RT-PCR to detect EpCAM, MUC1, CA125, and ERCC1 positive cells has also been reported [78].

Recently, circulating tumour DNA (ctDNA), was found in plasma, and demonstrated a high correlation with EOC prognosis. PCR-based approaches have been successfully applied in ctDNA analysis, but are limited to detection of certain specific known mutations. NGS has been used for DNA mutation profiling and tumour mutation burden determination, whilst other approaches, such as whole-genome sequencing and cancer-personalized profiling by deep sequencing, have a broad range of applications, including evaluation of tumour mutation burden, detection of epigenetic changes, and diagnostics or identification of resistance mutations in EOC [79].

## 4. Susceptibility to Prostate Cancer

### 4.1. BRCA1 and BRCA2 Genes in Prostate Cancer

The incidence of prostate cancer is greater among *BRCA* mutation carriers; namely, 1.35-fold and 2.64-fold greater in BRCA1 and BRCA2 carriers, respectively [80]. In the case of functional loss of *BRCA1* and *BRCA2* genes, DDR occurs by non-conservative and potentially mutagenic mechanisms. This genomic instability is considered to be related to the cancer predisposition caused by deleterious mutations in *BRCA* genes.

The incidence for *BRCA2* mutations is 4.45 (95% confidence interval (CI) 2.99–6.61), which is higher as compared to *BRCA1* (2.35, 95% CI 1.43–3.88) [81]. Consequently, *BRCA2* germline mutations increase the risk of prostate cancer 8.6-fold by the age of 65 years [82]. The first analysis of the IMPACT trial concluded that patients with *BRCA2* PVs have elevated levels of serum prostate specific antigen (PSA) at diagnosis, predominantly high Gleason tumours, increased rates of nodal and distant metastases, and finally high recurrence rates [83]. Three oligonucleotide/oligosaccharide-binding domains (OB) folds (OB1, 2, and 3) have been revealed in the DNA-binding domain (DBD) of BRCA2 by structural studies. OB1 and OB2 are associated with the highest risk of prostate cancer [84]. In a study of 6500 patients with *BRCA1* and *BRCA2* PVs, c.756-c 1000 and c.7914þ regions in BRCA2 were reported as negative biomarkers for high risk of Gleason 8b prostate cancer [85].

BRCA1 differentially regulates IGF-IR expression in androgen receptor (AR)-positive and AR-negative prostate cancer cells [86]. It has been reported that in a cohort of sporadic prostate cancer patients treated with radical prostatectomy, the higher probability of advanced tumour stage and the reduced disease-free survival were correlated with somatic *BRCA1* loss, which was due to hypermethylation or a deletion of the promoter [87].

Up to 8% of non-metastatic prostate cancer patients may respond to PARP inhibitors, irrespective of the fact that the HR deficiency is not derived from *BRCA* mutations [88]. This may be related to the *CDH1* gene loss (encodes cadherin 1) or inactivation of the *SPOP* gene (encodes Speckle-type POZ protein), which represent early events in carcinogenesis [89].

Strict separation of somatic and germline variants is not regularly performed; nevertheless, somatic *BRCA* mutations are more frequent in late stages of prostate cancer. Based on that, a new solid or liquid biopsy is highly recommended for an updated snapshot of the tumour.

### 4.2. Beyond BRCA

Prostate cancer is enriched for genomic alterations in DDR pathways [90]. In 2015, the Cancer Genome Atlas (TCGA) analyzed 333 primary prostate cancers. Alterations in DDR genes were common, affecting about one fifth of samples through mutations or deletions in *BRCA1/2*, *CDK12*, *ATM*, *FANCD2*, or *RAD51C* [91]. In terms of the metastatic setting, the study by the International Stand Up to Cancer-Prostate Cancer Foundation team (SU2C-PCF) evaluated 150 specimens and identified 8% with germline DDR mutations and 23% with somatic DDR alterations [50]. *BRCA2* was the most frequently mutated gene (13%), followed by *ATM* (7.3%), *MSH2* (2%) and *BRCA1*, *FANCA*, *MLH1*, *RAD51B* and *RAD51C* (0.3% for all). Similarly, in the large phase III PROfound study, among 28% of the analyzed samples were detected alterations in 15 DDR genes. The highest prevalence was found in *BRCA2* (8.7%), followed by *CDK12* (6.3%), *ATM* (5.9%), *CHEK2* (1.2%), and *BRCA1* (1%). The frequency of the DDR gene alterations between metastatic sites and primary disease was similar (32% vs. 27%, respectively) [92]. Co-occurring aberrations in two or more DDR genes were revealed in 2.2% of cases.

PROREPAIR-B is the first prospective multicenter cohort study evaluating the prevalence and effect of germline DDR mutations on metastatic castration-resistant prostate cancer outcomes [93]. Patients were screened for 107 DDR mutations. Among them, 16.2% were found to be DDR carriers, including 6.2% who carried mutations in *BRCA2*, *ATM*, or *BRCA1*.

The BRCAness phenotype may be induced pharmacologically or due to genetic alterations in HR genes other than *BRCA1/2*, including *ATM*, *ATR*, *CHEK1*, *RAD51*, the Fanconi anaemia complementation group family of genes and others. HR gene alterations were investigated in the TRITON2 study. The objective response rate (ORR) was 44% for patients with *BRCA* genes, but only 9.5% for *ATM* and 0% for *CDK12*, *CHEK2,* and other DDR genes [94]. Due to the doubts concerning BRCAness, experimental biomarkers have been proposed. Among them are included HR gene alteration, functional assays of HR capacity, as well as transcriptomic and mutational signatures [95].

### 4.3. Tumour Testing in Prostate Cancer

NGS implications are widely available for the detection of germline or somatic HR repair mutations, along with copy-number changes and genomic instability in prostate cancer [96]. The choice of the optimal material for genetic testing is challenging. Given the heterogeneity and instability of the tumour genome, metastatic sites may be better sources for identification of genetic alterations than the primary prostate tumour [97]. Brain and visceral metastases have the highest frequency of mutations among HR deficiency genes.

In the case of disease progression, it is recommended to repeat somatic mutation tests [98]. However, germline testing using NGS does not detect somatic mutations, which represent approximately 50% of *BRCA* mutations in metastatic castration-resistant prostate cancer [50]. For the detection of somatic HR repair mutations, CTC and ctDNA testing are highly recommended [99]. ctDNA plasma tests may be used in the absence of tissue or when re-biopsy is undesirable. However, the sensitivity of ctDNA plasma tests may be lower as compared to tumour tissue testing [100]. Ideally, a multiplex testing approach using different biological sample types can be implemented in order to increase the number of patients who undergo genomic analysis for actionable mutations.

From the screening programmes point of view, the previously reported IMPACT study facilitated annual PSA screening in families with germline *BRCA1/2* mutations [83,101]. The interim analysis of the study has reported that after 3 years of PSA screening in men with germline *BRCA1/2* mutations, those with a *BRCA2* mutation had increased incidence of prostate cancer, younger age at diagnosis, and higher risk of developing clinically significant tumours [83]. The study suggested that PSA testing in men with germline *BRCA1/2* mutations, with a threshold of 3ng/mL for biopsy, may be highly specific for the detection of early-stage disease [101]; nevertheless, this is not yet incorporated in compendium guidelines.

The NCCN guidelines (version 2.2021, July 2022) propose consideration of tumour testing for HR repair mutations (*BRCA1/2*, *ATM*, *CHEK2*, *PALB2*, *FANCA*, *RAD51D*, and *CDK12*), as well as microsatellite instability or MMR status (*MLH1*, *MSH2*, *MSH6*, and *PMS2*) in patients with regional, high-risk localised or metastatic prostate cancer [102]. The recommendation for germline testing also includes families with a history of cancer, and specific populations with high risk of prostate cancer, such as Ashkenazi Jews [102].

## 5. PARP Inhibitors Development across Tumour Types

There is an urgent need to better understand how the genomic and epigenomic heterogeneity intrinsic to EOC is reflected at the protein level, and how this information could potentially lead to prolonged survival [103]. The PARP inhibitors are a family of enzymes capable of catalyzing the transfer of ADP-ribose to target proteins. Among the 17 identified members of the PARP family, PARP-1 is the best characterized. It is responsible for approximately 90% of PARylation activity, whereas PARP-2 and to a lesser extent PARP-3 function in fewer, but overlapping, DNA repair processes [104]. With the binding of PARP to damaged sites, its catalytic activity and eventual release from DNA potentiate the response of a cancer cell to DNA breaks induced by chemotherapeutics and radiation [105]. The approved PARP inhibitors inhibit both PARP-1, -2, and -3. AZD5305 is a novel agent, designed as a highly potent and selective inhibitor of PARP-1 with DNA-trapping activity. The phase I/II PETRA trial evaluated AZD5305 as monotherapy in patients with advanced metastatic breast, pancreatic, or prostate cancer with germline *BRCA1*, *BRCA2*, *PALB2*, or *RAD51C* mutations [106]. There was preliminary evidence of early circulating tumour DNA responses. AZD5305 significantly improved pharmacokinetics and exposure to a target compared with the already approved first-generation PARP inhibitors, and thus represents a major advance over them.

Several PARP inhibitors in clinical development have different potencies as PARP-1 catalytic inhibitors and as PARP-‘trappers’. PARP inhibitors differ in terms of their metabolism; olaparib and rucaparib are metabolized by cytochrome P450 enzymes, whilst niraparib by carboxylesterase-catalyzed amide hydrolysis [107]. The potent antitumour effects of PARP inhibitors were originally observed in tumours harboring germline *BRCA1/2* mutations, such as familial breast and ovarian cancer. Among evaluated PARP inhibitors, olaparib, niraparib, and rucaparib are approximately 100-fold more potent than veliparib, while talazoparib has the most enhanced trapping potency [108]. The most common adverse events induced by PARP inhibitors are gastro-intestinal manifestations, myelosuppression, and fatigue. Nausea is the most prevalent gastro-intestinal adverse event. Symptoms are mainly mild and daily prokinetic, and antihistamine drugs are therapeutically recommended. Recalcitrant nausea or vomiting can be successfully controlled with a variety of antiemetic drugs, such as metoclopramide, prochlor-perazine, phenothiazine, dexamethasone, olanzapine, haloperidol, or lorazepam. Of note, the neurokinin-1 receptor antagonist aprepitant is contraindicated with olaparib, since it is a strong CYP3A4 inhibitor and may derange olaparib’s plasma concentrations. Other frequent gastrointestinal symptoms are constipation, vomiting, and diarrhoea, but grade 3 or 4 toxicities occur in less than 4% of patients. The treatment of choice is senna or polyethylene glycol 3350 for constipation, or loperamide for diarrhoea. Haematological toxicities tend to occur early after treatment initiation, with recovery after a few months. Among them, anaemia is the most common, related to PARP2 inhibition and erythrogenesis. In patients treated with niraparib, haematological adverse events represent the majority of grade 3 and 4 events, followed by rucaparib and olaparib. Haematological toxicities are the most common cause of dose modification, interruption, and discontinuation. The indications for transfusions include the symptomatic anaemia and the haemoglobin values of less than 7 g/dL. Thrombocytopenia of any grade is also more pronounced with niraparib. The cause of thrombocytopenia has been shown to be associated with a reversible decrease in megakaryocyte proliferation and maturation. Finally, fatigue is common for all PARP inhibitors and seems to be a class effect. Approximately 60–70% of patients experience fatigue of any grade with the three approved PARP inhibitors. The recommended management includes non-pharmacological approaches, such as exercise, massage therapy, and cognitive behavioural therapy, whilst pharmacological interventions with psychostimulants, such as methylphenidate and ginseng, may be considered in more symptomatic patients. The synthetic lethality may act against severe PARP inhibitor-mediated toxicity.

The successful story of PARP inhibitors in BRCA-deficient advanced breast and ovarian cancer has led to further investigation of their efficacy in prostate cancer, pancreatic and biliary tract malignancies, glioblastoma, and lung cancer. PARP inhibitors may also be effective in malignancies involving somatic mutations in DDR genes beyond *BRCA1/2*. They could also potentiate immunotherapeutic activity in many ways. Indeed, they increase neoantigen burden through DNA damage. Presence of HR deficiencies such as *BRCA1/2* mutations cause amplification of tumour mutational burden and contribute to immune checkpoint inhibitor sensitivity. Furthermore, PARP inhibitor-induced DNA damage could promote recruitment of T cells via the stimulator of interferon genes (STING) pathway and type I interferons. Finally, PARP inhibitors can lead to acute inflammation, remodeling of the tumour microenvironment, and thus enhancement of immune response [109].

### 5.1. Development of PARP Inhibitors in EOC

The standard treatment for ovarian cancer consists of cytoreductive surgery, followed by postoperative platinum-based chemotherapy. Neoadjuvant chemotherapy is an alternative option for selected patients, which offers the opportunity to test upfront chemosensitivity and to identify patients at higher risk of relapse [110]. Nevertheless, disease recurrence is a common phenomenon. Bevacizumab—a humanized monoclonal IgG antibody that targets vascular endothelial growth factor (VEGF) receptor—was the first antiangiogenic agent to show clear therapeutic activity in recurrent disease in combination with chemotherapy, based on the results of two randomized controlled phase III trials [111]. Clinical trials of PARP inhibitors have assessed their efficacy and tolerance in the treatment of EOC. Three PARP inhibitors have been approved for the management of EOC in different settings; olaparib, rucaparib, and niraparib.

Chronologically, in 2014, the EMA approved olaparib in maintenance setting for patients with recurrent high grade serous EOC and *BRCA1/2* mutations. The initial study enrolled 19 patients with platinum-sensitive relapse. This study demonstrated improved PFS vs. placebo (8.4 vs. 4.8 months, hazard ratio (HR) 0.35), which was more pronounced in the subset with germline/somatic *BRCA1/2* mutations (11.2 vs. 4.3 months, HR 0.18) [112]. In the same year, the FDA approved olaparib as the first-in-class PARP inhibitor for germline BRCA-mutated patients, previously treated with at least three lines of chemotherapy [113]. In 2018, the approval was expanded to all platinum-sensitive patients, regardless of *BRCA1/2* status. The confirmatory phase III SOLO-2 trial demonstrated median PFS of 19.1 vs. 5.5 months for olaparib and placebo, respectively, in germline BRCA1/2 mutants [114].

Rucaparib was approved by FDA and EMA in December 2016 and May 2018, respectively, for those previously treated with two or more lines of platinum-based chemotherapy, who cannot tolerate further platinum. The phase II ARIEL2 study confirmed that rucaparib prolonged PFS in patients with platinum-sensitive recurrence [115]. BRCA1/2-mutant cancers had improved response (80% vs. 10%) and prolonged PFS compared to the LOH low subgroup (HR 0.27, *p* < 0.0001). A subsequent post hoc analysis concluded that a cut off of 16% compared to 14% for the LOH assay may represent a better predictor of PFS [116].

Finally, the FDA and EMA approved niraparib in maintenance setting in March and November 2017, respectively, based on the phase III NOVA trial [117]. Patients with platinum-sensitive disease were enrolled, regardless of either germline BRCA1/2 or HR deficiency status, while results were stratified to investigate the potential predictive role of HR deficiency biomarkers. Definition of HR deficiency was determined by the myChoice HRD test, which incorporates LOH, telomeric allelic imbalance (TAI), and large-scale state transitions (LST). Median PFS for the non-germline BRCA carriers but signature-positive patients favoured niraparib (12.9 vs. 3.8 months, *p* < 0.001). Even patients without the HR-related signature achieved longer median PFS (6.9 vs. 3.8, *p* = 0.02). These data support that overall platinum-sensitivity status is correlated with PARP inhibitor sensitivity, although more benefit is seen in patients with canonical HR defects. A recently published meta-analysis explored the diversity of efficacy and safety of different PARP inhibitors in patients with EOC [118]. The results showed that either olaparib, niraparib, or rucaparib could prolong PFS over a placebo, whereas their long-term benefit was not limited to *BRCA* mutation status. Nevertheless, the analysis indicated that there was no difference in OS between olaparib and niraparib vs. the placebo. Finally, olaparib had the fewest grade 3 or higher adverse events, whereas no difference was identified between niraparib and rucaparib. However, we must be careful when considering those interpretations due to the methodological heterogeneity of the analysis.

Registration studies that led to approvals of PARP inhibitors for treatment of EOC are resumed in Table 1.

### 5.2. Development of PARP Inhibitors in Prostate Cancer

Until 2010, patients with metastatic castration-resistant prostate cancer have been treated with chemotherapy, which can be combined with androgen deprivation therapy (ADT). The addition of ADT to localised prostate radiotherapy improves survival as it sensitises prostate cancer to radiotherapy-induced cell death [126]. Technological advancements in the past two decades revealed that residual androgens, ADT-induced AR splice variants, and AR mutations are common mechanisms of metastatic castration-resistant prostate cancer. Within this context, AR signaling inhibitors are included among the agents that have been approved for the treatment of metastatic castration-resistant prostate cancer [127]. AR is a critical regulator of DDR in prostate cancer, through regulation of the expression and activity of DNAPK. This is an enzyme that is key for the process of repairing DSB through NHEJ and also serves as a transcriptional modulator. AR-induced DNAPK activation promotes transcriptional networks that lead to cell migration and metastasis, thus linking the AR-DNA repair axis to tumour progression [128]. The combination of PARP inhibition and AR signaling inhibitors could represent an example of synthetic lethality. AR is a ligand-inducible transcription factor, whereas AR signaling inhibitors cause HR deficit. ADT results in the state of BRCAness, leading to sensitivity of prostate cancer to PARP inhibition in combination with AR signaling inhibitors [129]. Multiple clinical trials are studying PARP inhibitors as either monotherapy or combined therapy for prostate cancer. Among them, olaparib was the first PARP inhibitor showing efficacy in metastatic castration-resistant prostate cancer patients with prior progression to standard treatment. The combination of rucaparib with AR has been approved to guide therapy based on paclitaxel harmful *BRCA* mutations in patients with metastatic castration-resistant prostate cancer. This is the rationale behind the clinical trials of veliparib and talazoparib as well. Key clinical trial data for these four PARP inhibitors in prostate are depicted in Table 2.

The United Kingdom (UK)-based TOPARP (Trial of PARP inhibition in prostate cancer) phase II trial was conducted in two stages. TOPARP-A assessed anti-tumour activity of olaparib in a sporadic metastatic castration-resistant prostate cancer population, whilst TOPARP-B was conducted in a subset with known genomic background, specifically *BRCA2* or *ATM* mutations [130,131]. In the TOPARP-A study, olaparib led to a response rate of 33% (95% CI 20–48), reduction in CTC of 29%, and 50% decrease in PSA levels of 22% over the whole cohort [130]. However, when TOPARP-B patients were stratified based on NGS results, 88% responded to olaparib; namely, 80% of those with *ATM* mutations and all BRCA2 mutants. On the other hand, only 2 of 33 biomarker-negative patients (6%) had a response to olaparib (sensitivity of 88% and specificity of 94%) [131]. These studies concluded that olaparib is primarily effective in metastatic castration-resistant prostate cancer patients with HR deficiency. Tumours with *BRCA1* or *BRCA2* alterations were more sensitive to olaparib as compared to those with alterations in any other DDR gene.

In the phase III biomarker-driven PROfound trial, the patients were divided into two cohorts. Cohort A assigned patients with *BRCA1*, *BRCA2*, and *ATM* mutations, and cohort B comprised those with mutations in one of the remaining 12 DDR genes [132]. The patients were given olaparib 300 mg twice daily and second line AR signaling inhibitors in a 2:1 ratio. In cohort A, the median radiographic PFS was 7.4 and 3.5 months in favour of olaparib, whilst the median OS was 18.5 and 15.1 months, respectively (HR 0.64, *p* = 0.02). The study met the primary endpoint for radiographic PFS. Based on the positive results of the PROfound trial, the FDA approved olaparib in January 2020 for the treatment of metastatic castration-resistant prostate cancer in patients with deleterious DDR gene mutations, followed by new hormone therapy. Even though it is an approved modality in the United States of America and Europe, this is not the case in the UK.

The TRITON2 and GALAHAD phase II trials investigated the potential therapeutic benefit of rucaparib and niraparib, respectively, in metastatic castration-resistant prostate cancer patients with DDR mutations and disease progression after AR signalling inhibitor or chemotherapy [133,135]. The TRITON2 trial enrolled 190 metastatic castration-resistant prostate cancer patients to be treated with rupacarib 600 mg twice daily. Among them, 52% had a *BRCA1/2* mutation, and the remaining had *ATM* (30%), *CDK12* (7%), *CHEK2* (4%), and other mutated genes (7%). The ORR was 44% for patients with *BRCA* mutations, but only 9.5% for *ATM,* and 0% for the remaining DDR genes [133]. These positive preliminary findings led to the FDA approval of rucaparib in May 2020 for *BRCA1/2* mutated metastatic castration-resistant prostate cancer patients who progressed after one to two lines of AR-directed therapy and one taxane-based chemotherapy. However, the TRITON2 study has not detected accurate biomarkers in non-*BRCA*-mutated tumours.

The GALAHAD trial enrolled 165 metastatic castration-resistant prostate cancer patients with germline pathogenic or somatic biallelic pathogenic alterations in *BRCA1* or *BRCA2* (BRCA cohort), or in other prespecified DDR genes (non-BRCA cohort), who were treated with niraparib 300 mg twice daily. The composite response rate—defined as ORR, conversion of CTC to <5/7.5 mL blood or ≥50% decline in PSA—was 63% in the BRCA and 17% in the non-BRCA cohort, respectively [135]. Similar to olaparib, rucaparib was approved by the FDA—but not by the EMA—for the treatment of metastatic castration-resistant prostate cancer patients with germline and/or somatic *BRCA1/2* mutations, who progressed on AR signaling inhibitor or taxane. Of note, the GALAHAD study stratified patients with biallelic mutations, whilst the TRITON2 and PROfound trials evaluated mono- and biallelic mutations in tumour tissue or plasma and tumour tissue, respectively. Whether the origin and type of *BRCA1/2* mutation (monoallelic vs. biallelic, somatic vs. germline) may potentially affect therapeutic response to PARP inhibitors requires further investigation.

## 6. Developing Predictive Biomarkers for PARP Inhibitors

The first clinical biomarker for the evaluation of response to PARP inhibitors was platinum sensitivity. The platinum-free interval is correlated with the clinical benefit rate of olaparib in *BRCA1/2* mutated EOC patients. The reported—in a phase I study—clinical benefit rate for the olaparib were 69.2% and 45.8% for the platinum-sensitive and platinum-resistant groups, respectively [138]. The subset of patients with germline *BRCA1/2*-mutated, platinum-sensitive disease achieved the best response to olaparib. On the other hand, the response to platinum-based chemotherapy is not always compatible with the response to PARP inhibitors. This is based on the fact that platinum sensitivity may result from defects in other DDR mechanisms, including NER [139]. Moreover, the secondary restoration of the function of *BRCA1/2* or other HR genes may lead to resistance to PARP inhibition, rather than to platinum resistance [140].

Multiplexed NGS panels investigate the mutation status of multiple genomic regions of interest, either through amplification or capture-based technologies. Multiplexed panels are successfully implemented in clinical practice, based on their lower cost and burden of bioinformatics requirements for the analysis of the data.

Molecular signatures, such as the HR deficiency scores, are crucial for therapeutic decisions. Most of the evidence on the predictive value of such signatures was obtained from the randomized trials of PARP inhibitors rucaparib and niraparib in EOC. HR deficiency is involved in the tumourigenesis of approximately 50% of high-grade serous ovarian carcinoma, whilst about 20% are caused by mutations in HR genes beyond *BRCA1/2* [141].

Several FDA-approved companion diagnostic tests for PARP inhibitors are currently available. BRACAnalysis CDx consists of two in vitro assays for germline *BRCA1/2* mutational identification; the BRACAnalysis CDx Sanger sequencing and the BRACAnalysis CDx Large Rearrangement Test (BART^®^). They are used for sequence variants and large rearrangements, respectively. Potential limitations of BRACAnalysis CDx are the detection of deletions > 5 bp, insertions > 2 bp, RNA transcript processing errors, and differentiation between gene duplication and triplication [142]. An additional critical limitation of these signatures is that the mutational/LOH patterns do not revert when a tumour has recovered HR function. As such, they may not be able to accurately predict PARP inhibitors’ sensitivity in the subset of patients who have been previously treated and progressed on DNA damaging chemotherapy. Myriad’s myChoice HR deficiency is an enhancement of BRACAnalysis CDx that identifies both germline and somatic *BRCA1/2* mutations, along with HR deficiency [143]. The created genomic scarring composite score represents a sum of LOH, TAI, and LST.

The RAD51 assay is also a promising candidate for predicting responses to PARP inhibition. RAD51 is an important protein in the HR repair pathway that can be easily detected with an immunofluorescence assay [144]. The induction of RAD51 foci formation after DNA damage has been associated with HR repair proficiency [139]. RAD51 can accurately identify all *PALB2*-mutated tumours as HR-deficient in clinical breast samples [145]. The RAD51 foci assay has also successfully been used as an in vitro predictive biomarker for PARP inhibition in cultures from the ascitic fluid of patients with EOC [146].

As far as prostate cancer is concerned, it has been reported that 30% of patients with metastatic castration-resistant prostate cancer respond to treatment with PARP inhibitors [131]. The first successful prostate cancer biomarker study was the previously mentioned PROfound study, which demonstrated that patients with *BRCA1*, *BRCA2*, and *ATM* alterations responded better to PARP inhibitors and achieved a longer radiographic PFS and OS. In contrast, patients with long-tail DDR alterations did not experience clinical benefit [92]. Moreover, prostate cancer with *BRCA2* had better outcome as compared to those with *BRCA1* mutations, after treatment with PARP inhibitors [147]. Furthermore, Lotan et al., reported that in a three-cohort study, patients with primary prostate cancer and germline *BRCA2* mutations had the highest genomic scarring composite score, followed by the *ATM* and *CHEK2* alterations [148]. Apparently, those with *BRCA2* mutations respond better to PARP inhibitors as compared to the prostate cancer patients with *ATM* and *CHEK2* alterations [132]; nevertheless, the same correlation with higher genomic scarring composite scores has been revealed in the respective DDR gene mutations [148]. The implication of PARP inhibitors beyond *BRCA1/2* mutation—in cases of the ‘BRCAness’ phenotype—highlights the importance of future trials investigating predictive biomarkers beyond BRCA [149].

The activity of PARP1 is believed to be a new biomarker for sensitivity to PARP inhibitor, as it has been reported that increased PARP1 activity correlates positively with disease progression in prostate cancer. PARP1 enhances E2F1-related mechanisms of HR [150]. E2F1 is a transcription factor that regulates the cell cycle and activates cell proliferation. Therefore, the inhibition of PARP1 results in BRCAness, due to decreased expression of DDR genes.

Finally, a recent study used CRISPR-Cas9 screens for the potential identification of PARP inhibitors’ sensitivity marker. Interestingly, it has been revealed that alterations in the genes encoding the RNase H2 enzyme complex (RNASEH2A, RNASEH2B, and RNASEH2C) may cause PARP inhibitor sensitivity through impaired ribonucleotide excision repair [151].

## 7. BRCA Mutations and Radiation Response

Mutations in genes implied in response to DNA damage were shown to impact on radiation response in various preclinical models. Indeed, the NHEJ and HR are two major mechanisms required for repair of radiation-induced DSBs [152]. In vitro and in vivo experiments demonstrated increased sensitivity to ionizing radiation in ovarian cancer cells carrying defective BRCA1, with data suggesting a role of BRCA1 in Foxp3 mediated radiation resistance [152]. There are therefore theoretical concerns on potential increased radiation sensitivity of normal tissue among *BRCA1* mutation carriers, but also potential increased effectiveness against tumours. Despite this preclinical background, clinical data, mainly obtained in breast cancer patients, did not provide a clear signal that there would be differences in prognosis after adjuvant radiotherapy in patients with BRCA-associated breast cancer or sporadic breast cancer [153]. The place of radiotherapy for ovarian cancers is now quite limited, though the survival benefit afforded by molecular targeted agents leads to long-term survivors, with new indications for stereotactic body radiotherapy in oligoprogressive or oligopersistent disease. For prostate cancer, radiation therapy has a more substantial role, especially in curative strategies. There is a strong rationale to associate radiotherapy with PARP inhibitors, and preclinical data confirmed the potential of such association, leading to more frequent DNA damages, but also to immunogenic effects (e.g., enhanced infiltration of cytotoxic T lymphocytes into the tumour bed, increased expression of PD-1/PDL-1) [154]. To date, only few early phase clinical trials tested PARP inhibitors with radiotherapy, showing the feasibility of such association [155]. Howeve, it remains uncertain whether such an association would lead to different efficacy or safety profiles among patients with *BRCA1/2* mutations. The possibility to reverse systemic resistance to immunotherapy or to PARP inhibitors through irradiation of selected metastatic sites is another area of research [156].

## 8. Conclusions and Future Directions

Copy number variations (CNVs) which include deletions, duplications, inversions, translocations, and other forms of chromosomal re-arrangements are common to human cancers [157,158,159,160]. Apart from the local chromosomal architecture, CNVs are driven by the internal cellular or nuclear physiology of each cancer tissue [161]. A recently published study proposed GraphChrom—a novel graph neural network-based framework—for predicting cancer from chromosomal rearrangement endpoints [162]. Approximately half of all cancers have somatic integrations of retrotransposons. A study analyzed the patterns and mechanisms of cancer retrotransposition on a multidimensional scale, across 2954 cancer genomes, integrated with rearrangement, and transcriptomic and copy number data [163]. Major restructuring of cancer genomes may emerge from aberrant L1 retrotransposition events in tumours with high retrotransposition rates. L1-mediated deletions can promote the loss of megabase-scale regions of a chromosome, which may involve centromeres and telomeres. The majority of such genomic rearrangements would be harmful for a cancer clone. However, L1-mediated deletions may promote cancer-driving rearrangements that involve the loss of tumour-suppressor genes and/or the amplification of oncogenes. Through that mechanism, cancer clones acquire new mutations that help them to survive.

*BRCA1* and *BRCA2* genes encode proteins required to restore broken DNA by HR. If mutations inactivate either the *BRCA1* or *BRCA2* gene, then the broken DNA can become pathogenic. Pieces of DNA get lost or reattach at the wrong positions on the original or different chromosomes. In *BRCA1* or *BRCA2* mutants, these errors lead to chromosomal re-arrangements and shifts typical of hereditary breast cancers. Chromosomal re-arrangements may be critical events leading to hereditary breast cancers, but our knowledge of what causes these events is limited. Immune deficits in *BRCA1* and *BRCA2* mutants may allow the reactivation of latent EBV infections or new herpes viral infections [164]. DNA breaks induced by exogenous human herpesvirus 4 (EBV) nucleases may then become pathogenic. The availability of breast cancer genomic sequences allows testing of the possibility that EBV contributes to the incorrect reattachment of broken chromosomes in hereditary breast cancer. The relationship between breast cancer chromosome breaks and viruses may become actionable.

Based on the available evidence on germline and tumour testing for EOC patients, germline genetic testing should be offered to all women diagnosed with EOC. The analysis should be able to detect damaging variants in all genes associated to ovarian cancer susceptibility, rather than just *BRCA1/2* genes. Tumour testing, at least for *BRCA1/2* genes, is recommended for all women testing negative for germline PVs. PARP inhibitors have attracted great attention and illustrate a paradigm of bench-to-bedside medicine. HR deficiency remains a strong predictor of clinical benefit from these agents. The current state of HR deficiency testing can identify patients with EOC who will most likely benefit from PARP inhibitors. Precise biomarkers for negative response prediction are pivotal. Better understanding of BRCA and its role in the development and outcomes of EOC provides a great potential to prevent many cases through improved access to genetic screening, and also to revolutionize the long-term treatment.

Equally, prostate cancer germline and somatic mutations have been found especially in the *BRCA* genes, and subsequently, germline and somatic testing is recommended. This has changed the molecular classification of prostate cancer and expands the available therapeutic options. The evaluation of the safety and efficacy of multiple PARP inhibitors has led to encouraging results. This is crucial as prostate cancer was devoid of predictive therapeutic biomarkers in the past. PARP inhibitors were initially thought to be relevant for DDR mutations, but their therapeutic implication has been expanded, as they may be combined with AR signaling and immune checkpoint inhibitors. The assessment of this strategy with potential for an increased targeted population represents an ongoing effort. The clinical outcome can be affected by several parameters including tumour vs. liquid biopsy, somatic vs. germline mutations, and programmed death-ligand 1 (PD-L1) positivity on tumour cells vs. immune cells. Finally, it is prudent to explore the resistance mechanisms to PARP inhibitors by utilizing non-invasive tools such as cfDNA, as this would help the development of subsequent treatment strategies.

## Figures and Tables

**Figure 1 cancers-14-03888-f001:**
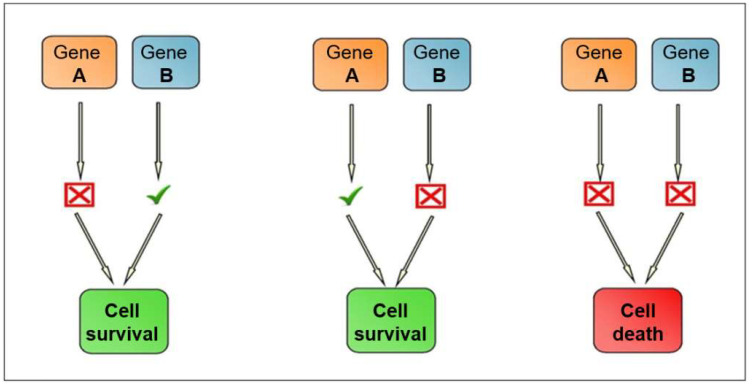
Schematic of synthetic lethality in cancer.

**Figure 2 cancers-14-03888-f002:**
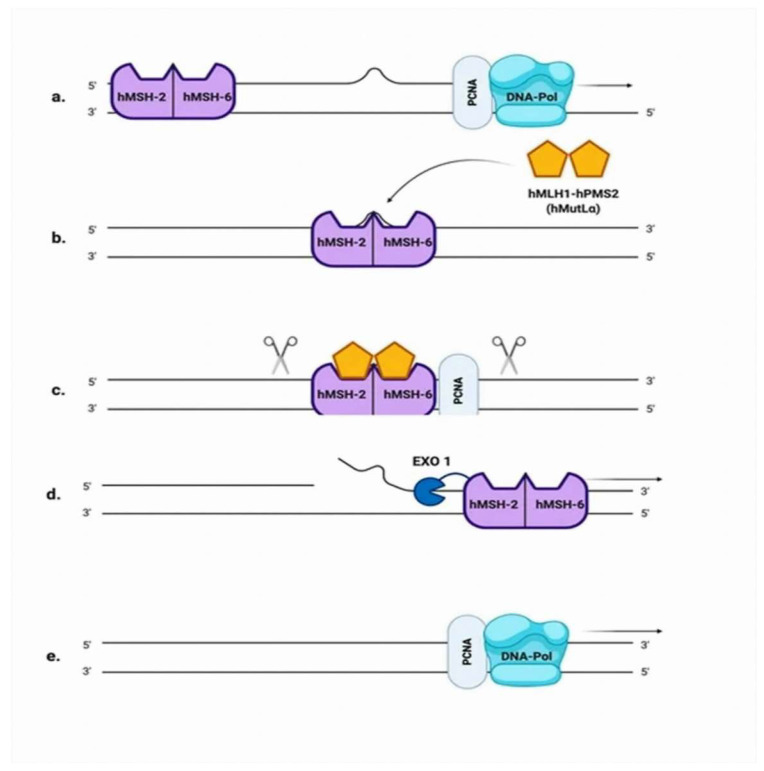
Schematic representation of the mismatch repair (MMR) system. (**a**) MMR enzymes scan the DNA and remove the wrongly incorporated bases from the newly synthesized, non-methylated strand by using the DNA polymerase. (**b**) In MMR, the incorrectly added base is detected after replication by hMutSα, which recruits hMutLα. (**c**) hMutLα detects this base and removes it from the newly synthesized strand. (**d**) hMutSα activates EXO1 and the entire segment of DNA is removed. (**e**) DNA polymerase participates on the replacement of the DNA by correctly paired nucleotides.

**Figure 3 cancers-14-03888-f003:**
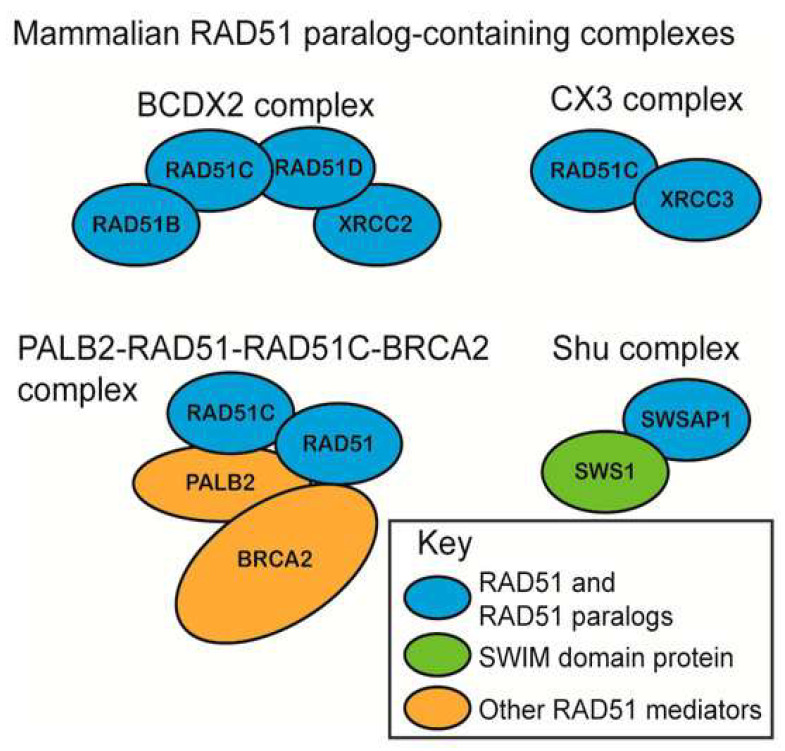
Schematic representation of the mammalian RAD51 paralog-containing complexes.

**Table 1 cancers-14-03888-t001:** Clinical trials of PARP inhibitors in ovarian cancer.

Study	Phase	Population	Treatment Arms	Outcome	P	Ref
STUDY 19	II	(1) Platinum-sensitive, advanced HGSOC (2) At least two prior lines of platinum-based CTH (3) Unselected for BRCA status	(A) Olaparib 400 mg BID (B) Placebo	(A): Median PFS 1. Overall population: 8.4 vs. 4.8 m 2. BRCA mutants: 11.2 vs. 4.3 m 3. BRCA wild type: 7.4 vs. 5.5 m (B): OS 1. Overall population: 29.8 vs. 27.8 m 2. BRCA mutants: 34.9 vs. 31.9 m 3. BRCA wild type: 24.5 vs. 26.2 m (C): ORR 12% vs. 4%	(A1): <0.001 (A2): <0.0001 (A3): 0.0075 (B1): 0.44 (B2): 0.19 (B3): 0.96 (C): 0.12	[112]
STUDY 42	II	(1) Platinum-resistant, advanced HGSOC (2) BRCA mutations	Olaparib 400 mg BID	(1) ORR: 34% (2) MDR: 7.9 m (3) PFS: 7 m (4) OS: 16.6 m		[113]
SOLO 2	III	(1) Platinum-sensitive, advanced HGSOC or HGEOC (2) At least two prior lines of platinum-based CTH (3) BRCA mutations	(A) Olaparib 300 mg BID (B) Placebo	Median PFS: 19.1 vs. 5.5 m	<0.0001	[114]
ARIEL2	II	Platinum-sensitive, advanced HGSOC or HGEOC	Rucaparib 600 mg BID	(A): Median PFS 1. BRCA mutants: 12.8 m 2. BRCA wild type LOH high: 5.7 m 3. BRCA wild type LOH low: 5.2 m (B): ORR 1. BRCA mutants: 80% 2. BRCA wild type LOH high: 39% 3. BRCA wild type LOH low: 13%	(A1): <0.0001 (A2): 0.011 (A3): 0.011	[115]
NOVA	III	(1) Platinum-sensitive, advanced HGSOC (2) At least two prior lines of platinum-based CTH (3) Stratification by gBRCAmut	(A) Niraparib 300 mg BID (B) Placebo	Median PFS (1) gBRCA mutants: 21 vs. 5.5 m (2) BRCA wild type HRD (+): 12.9 vs. 3.8 m (3) Overall non-gBRCA mutants: 9.3 vs. 3.9 m	(1): <0.0001 (2): <0.00001 (3): <0.0001	[117]
STUDY 10	I/II	(1) Platinum-sensitive, advanced HGSOC or HGEOC; (2) gBRCAmut (phase II PART 2A)	Rucaparib 600 mg BID	(1) ORR: 59.5% (2) MDR: 7.8 m		[119]
SOLO 1	III	(1) Platinum-sensitive, advanced HGSOC (2) BRCA mutations	(A) Olaparib 300 mg BID (B) Placebo	Median PFS: NR vs. 13.8 m 3-year PFS: 69% vs. 35%	<0.001 <0.001	[120]
SOLO 3	III	Recurrent gBRCAm EOC	(A) Olaparib (B) CTH	Median PFS: 13.4 vs. 9.2 m	0.013	[121]
PRIMA	III	Newly diagnosed advanced EOC with response to platinum-based CTH	(A) Niraparib 300 mg BID (B) Placebo	Median PFS (1) HRD (+): 21.9 vs. 10.4 m (2) Overall population: 13.8 vs. 8.2 m	(1): <0.001 (2): <0.001	[122]
QUADRA	II	(1) Platinum-sensitive, advanced HGSOC (2) HRD (+)	Niraparib 300 mg BID	(1) ORR 27.5% (2) DCR 68.6%		[123]
ARIEL3	III	Recurrent EOC after response to platinum-based CTH	(A) Rucaparib 600 mg BID (B) Placebo	Median PFS (1) BRCA mutants: 16.6 vs. 5.4 m (2) HRD (+): 13.6 vs. 5.4 m (3) ITT population: 10.8 vs. 5.4 m	(1): <0.0001 (2): <0.0001 (3): <0.001	[124]
PAOLA-1	III	Newly diagnosed, advanced, high-grade ovarian cancer with response after first-line platinum-taxane CTH plus bevacizumab	(A) Bevacizumab + olaparib maintenance (B) Bevacizumab + placebo	Median PFS (1) Overall population: 22.1 vs. 16.6 m (2) HRD (+): 37.2 vs. 17.7 m (3) HRD without BRCA mutations: 28.1 vs. 16.6 m	(1): <0.001	[125]

Abbreviations: PARP: poly(ADP-ribose) polymerase; Ref: reference; HGSOC: high-grade serous ovarian cancer; HGEOC: high-grade endometrioid cancer; gBRCAmut: germline BRCA mutation; BID: twice a day (bis in die); ORR: overall response rate; MDR: median duration of response; m: months; CTH: chemotherapy; PFS: progression-free survival; OS: overall survival; NR: not reached; EOC: epithelial ovarian cancer; HRD: homologous recombination deficiency; DCR, disease control rate; LOH: loss of heterozygosis; ITT: intent-to-treat.

**Table 2 cancers-14-03888-t002:** Clinical trials of PARP inhibitors in prostate cancer.

Clinical Trial ID	Phase	PARP Inhibitor	Population	PSA Response Rate	Primary Endpoint	Ref
NCT01682772	II	Olaparib	mCRPC patients previously treated with abiraterone or enzalutamide, and cabazitaxel	33% of patients (95%, 20–48)	RR, PSA, CTC	[130]
NCT01682772	II	Olaparib	mCRPC patients: (1) previously treated with one or two taxanes (2) DDR gene mutations	PSA levels decrease by ≥ 50%: 100% of BRCA2 and FANCA mutated mCRPC patients	RR, PSA, CTC	[131]
NCT02987543	III	Olaparib	mCRPC patients: (1) disease progression whilst on enzalutamide or abiraterone (2) ≥1 HRR gene mutation	Olaparib group: 30% of patients Control group: 10% of patients	rPFS	[132]
NCT02952534	II	Rucaparib	mCRPC patients: germline or somatic alteration in ≥1 prespecified HRR gene	47.8% of BRCA-mutated patients (95%, 26.8–69.4)	ORR	[133]
NCT04455750	III	Rucaparib	mCRPC patients, resistant to testosterone-deprivation therapy	Not completed	rPFS, OS	[134]
NCT02854436	II	Niraparib	mCRPC patients: (1) DDR gene mutations (2) disease progression on taxane and AR-targeted therapy	57% of patients (95% CI, 34–77)	ORR	[135]
NCT03148795	II	Talazoparib	mCRPC patients: (1) DDR-mutated (2) disease progression on taxane or AR-targeted therapy	Not completed	ORR	[136]
NCT04821622	III	Talazoparib	mCSPC patients with DDR gene mutations	Not completed	rPFS	[137]

Abbreviations: PARP: poly(ADP-ribose) polymerase; Ref: reference; mCRPC: metastatic castration-resistant prostate cancer; RR: response rate; PSA: prostate specific antigen; CTC: circulating tumour cells; DDR: DNA damage repair; HRR: homologous recombination repair; rPFS: radiographic progression-free survival; ORR: objective response rate; OS: overall survival; AR: androgen receptor; mCSPC: metastatic castration-sensitive prostate cancer.

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
