# Peer review of "BRCA Mutations in Ovarian and Prostate Cancer: Bench to Bedside"

_cancers, 2022, doi:10.3390/cancers14163888_

Round 1

Reviewer 1 Report

Timely overview on the importance of BRCA mutations in ovarian and prostate cancer.

Here a few points to potentially consider:

PAGE 1

Simple summary, line 6: … are common in prostate cancer as well. BRCA2 lesions are found in 12% of mCRPC, but very rarely in primary PC.

Line 10: PARP inhibitor approval is only in mCRPC with alterations in DNA repair genes.

Abstract line 10: Tumors with BRCA1 or BRCA2 mutations are highly …

PAGE 2

Abstract lines 2-4: Please revise.

Introduction line 19: That 90% of mCRPC harbor clinically actionable alterations remains to be proven.

Synthetic lethality, line 4: unclear, what is meant by second pair of the gene? Monoallelic inactivation of a DDR gene is usually sufficient to increase cancer risk.

 PAGE 5

As mentioned at the bottom of this page (and several times more after that), ATM alterations are nearly as frequent as BRCA2 alterations in mCRPC. However ATM inhibitors are mentioned nowhere in the review. It may be worth saying that results for the phase I dose-escalation study for M3541 were just published but unfortunately do not look promising (SN Waqar et al, Invest New Drugs, 2022, 40, 596). Concerning gynecological cancer patients, a clinical study combining the ATR inhibitor ceralasertib with olaparib has just been initiated (S Banerjee et al, Int J Gynecol Cancer, 2021, 31, 1471).  

PAGE 6

3., line 3: as stated above, ATM inhibitors would also be worth mentioning.

3.1, 2nd paragraph: The sentence “The identified through cascade …” is not understandable.

3.2, line 1: what is meant by magnitude?

PAGE 7

3.3: What about testing of CTCs or plasma DNA?

4.1: Giving the numbers would be helpful: 2.64-fold greater risk in BRCA2 carriers and 1.35-fold greater risk in BRCA1 carriers.

PAGE 8

Line 14: It is claimed that up to 8% of nmPC patients may respond to PARP inhibitor. Where does this number come from?

PAGE 9

Paragraph 5: Approved PARP inhibitors are pan-PARP. It would be worth mentioning the novel clinical PARP1-selective inhibitor AZ5305.

PAGE 10

Line 2: Only one sentence is devoted to side-effects, which is a bit short in view of the multiple clinical observations made. More precise data and management options for adverse events would be informative.

5.1: Why is the anti-angiogenesis agent bevacizumab not listed as standard treatment? It is approved since 2018.

PAGE 11

End of paragraph 5.1: it may be worth mentioning a meta-analysis reporting that the 3 approved PARP inhibitors have similar PFS impact on ovarian cancer, but that olaparib is better tolerated (J Luo et al, Front Oncol, 2022, 12:815265).

PAGE 14

5.2: mCRPC is treated with ADT and AR signaling inhibitors, and usually only later with chemotherapy.

Concerning AR signaling and DNA damage, please mention papers from K Knudsen who has done pioneering and extensive work in this area (e.g. The AR-DNA repair axis: insights into prostate cancer aggressiveness. Can J Urol, 2019, 26 (5 Suppl 2), 22-23).

Table 2: Phase III studies in mPC with rucaparib (NCT04455750) or talazoparib (NCT04821622) are not mentioned.

Author Response

Dear Editor and Reviewers,

I am pleased to resubmit for publication the revised version of cancers-1813589 manuscript, entitled “BRCA Mutations in Ovarian and Prostate Cancer: Bench to Bedside”.

Thankfully the reviewers provided us with a great deal of guidance, regarding how to better position the article. We are hopeful you agree that this revision will update our comprehensive review. All the comments have been addressed, as shown in the revised version of the manuscript, along with this point-by-point response to the reviewers' comments.

All corresponding are blue changes in the manuscript.

Reviewer #1:

  • General comment:

Timely overview on the importance of BRCA mutations in ovarian and prostate cancer.

Here a few points to potentially consider:”.

Response:

Thank you for your positive reinforcement and constructive feedback. We appreciate the opportunity to revise our work for consideration for publication.

  1. PAGE 1

Simple summary, line 6: … are common in prostate cancer as well. BRCA2 lesions are found in 12% of mCRPC, but very rarely in primary PC.

Line 10: PARP inhibitor approval is only in mCRPC with alterations in DNA repair genes.

Abstract line 10: Tumors with BRCA1 or BRCA2 mutations are highly ….

Response:

Thank you for your comments.

  • Simple summary, line 6: We have now added the sentence “Among them, BRCA2 lesions are found in 12% of metastatic castration-resistant prostate cancer, but very rarely in primary prostate cancer.”, as you kindly recommended.

  • Line 10: We fully agree and rephrased as per your advice (“Poly(ADP-ribose) polymerase (PARP) inhibitors are now a standard therapy for EOC patients and more recently have been approved for the metastatic castration-resistant prostate cancer with alterations in DDR genes.”).

  • Abstract line 10: We have now rephrased appropriately (“Tumors with BRCA1 or BRCA2 mutations are highly sensitive to PARP inhibitors.”).

  1. PAGE 2

Abstract lines 2-4: Please revise.

Introduction line 19: That 90% of mCRPC harbor clinically actionable alterations remains to be proven.

Synthetic lethality, line 4: unclear, what is meant by second pair of the gene? Monoallelic inactivation of a DDR gene is usually sufficient to increase cancer risk.

Response:

Thank you for your recommendations.

  • Abstract lines 2-4: We have revised as follows:

PARP inhibitors are in advanced stages of development as a treatment for metastatic castration-resistant prostate cancer. However, there is a major concern regarding the need to identify reliable biomarkers predictive of treatment response.”.

  • Introduction line 19: We agree and rephrased appropriately:

Approximately, 20% of castration-resistant prostate cancer patients harbour germline or somatic mutations in one of the DDR genes, which supports the mechanism of synthetic lethality [9].”.

  • Synthetic lethality, line 4: We have now rephrased for better clarity as follows:

The synthetic lethality strategy requires gene pairs that, when simultaneously inactivated, cause cell death.”.

  1. PAGE 5

As mentioned at the bottom of this page (and several times more after that), ATM alterations are nearly as frequent as BRCA2 alterations in mCRPC. However ATM inhibitors are mentioned nowhere in the review. It may be worth saying that results for the phase I dose-escalation study for M3541 were just published but unfortunately do not look promising (SN Waqar et al, Invest New Drugs, 2022, 40, 596). Concerning gynecological cancer patients, a clinical study combining the ATR inhibitor ceralasertib with olaparib has just been initiated (S Banerjee et al, Int J Gynecol Cancer, 2021, 31, 1471).

Response:

Thank you for your consideration that is of great value. We have now added the final part of the section “2.3. Mismatch repair (MMR)”, based on the recommended references (48-50).

  1. PAGE 6

3., line 3: as stated above, ATM inhibitors would also be worth mentioning.

3.1, 2nd paragraph: The sentence “The identified through cascade …” is not understandable.

3.2, line 1: what is meant by magnitude?

Response:

Thank you for your recommendations.

  • 3., line 3: We have added a few lines as you kindly recommended.

Moreover, mutated or downregulated ARID1A significantly compromises HR repair of DNA DSBs. ARID1A is recruited to DSBs via its interaction with ATR. ATR inhibition in ARID1A defective cells thus increases large scale genomic rearrangements and ultimately causes cell death. Loss of function mutations in ARID1A leading to a loss of protein expression are a frequent observation in EOC of clear cell and endometroid histology.”.

  • 3.1, 2nd paragraph: We have rephrased appropriately, as follows:

Cascade screening is the systematic identification and testing of relatives of a known mutation carrier. This strategy determines whether asymptomatic relatives also carry the known variant, in view of offering risk-reducing options to reduce the morbidity and mortality.”.

  • 3.2, line 1: We have now rephrased in order to make better sense:

nevertheless, hereditary ovarian cancer is most commonly caused by a mutation in BRCA1/2 genes.”.

  1. PAGE 7

3.3: What about testing of CTCs or plasma DNA?

4.1: Giving the numbers would be helpful: 2.64-fold greater risk in BRCA2 carriers and 1.35-fold greater risk in BRCA1 carriers.

Response:

Thank you for your recommendations.

  • 3.3: We have added the final two paragraphs of that section (References 74, 75 and 76).

  • 4.1: We have incorporated these numbers as you kindly advised.

  1. PAGE 8

Line 14: It is claimed that up to 8% of nmPC patients may respond to PARP inhibitor. Where does this number come from?

Response:

Thank you for your recommendations. We have added the relevant reference in the revised paper (Reference 85).

  1. PAGE 9

Paragraph 5: Approved PARP inhibitors are pan-PARP. It would be worth mentioning the novel clinical PARP1-selective inhibitor AZ5305.

Response:

Thank you for your recommendations. We have added few sentences about AZ5305 and the PETRA trial (Reference 103).

  1. PAGE 10

Line 2: Only one sentence is devoted to side-effects, which is a bit short in view of the multiple clinical observations made. More precise data and management options for adverse events would be informative.

5.1: Why is the anti-angiogenesis agent bevacizumab not listed as standard treatment? It is approved since 2018.

Response:

Thank you for your recommendations.

  • Line 2: We have now incorporated information about PARPs’ mediated side-effects and their management at the end of that paragraph.

  • 5.1: We have now incorporated a few words in terms of the therapeutic role of bevacizumab in the first paragraph of the section “5.1”, along with the reference 108.

  1. PAGE 11

End of paragraph 5.1: it may be worth mentioning a meta-analysis reporting that the 3 approved PARP inhibitors have similar PFS impact on ovarian cancer, but that olaparib is better tolerated (J Luo et al, Front Oncol, 2022, 12:815265).

Response:

Thank you for your recommendations. We discuss the results of the recently published meta-analysis at the end of the chapter “5.1” in the revised manuscript and incorporated the relevant reference as you kindly recommended (115).

  1. PAGE 14

5.2: mCRPC is treated with ADT and AR signaling inhibitors, and usually only later with chemotherapy.

Concerning AR signaling and DNA damage, please mention papers from K Knudsen who has done pioneering and extensive work in this area (e.g. The AR-DNA repair axis: insights into prostate cancer aggressiveness. Can J Urol, 2019, 26 (5 Suppl 2), 22-23).

Table 2: Phase III studies in mPC with rucaparib (NCT04455750) or talazoparib (NCT04821622) are not mentioned..

Response:

Thank you for your recommendations.

  • 5.2: We have rephrased the relevant part of the first paragraph, based on your consideration.

  • We have also cited the paper of KE Knudsen “The AR-DNA repair axis: insights into prostate cancer aggressiveness” and mentioned the AR-induced DNAPK activation that lead to progression and metastasis (Reference 125).

  • Table 2: We have added the two studies as you kindly recommended (References 131 and 134, respectively).

Reviewer 2 Report

In this review, the authors discuss mutations in BRCA1/2 and related DDR genes and their implications in cancer diagnosis and therapy. The review is relevant and captures some of the literature. However, there are many deficiencies which I point below. Often the authors are too simplistic in their discussion and/or ignore key findings in the field. As a DNA damage repair investigator, I think that this review reads somewhat superficial in this area. I recommend that the authors revise their manuscript in light of my comments below. It is my opinion that such revisions will significantly improve the manuscript and increase its impact.

Major comments

1.       Discussion starting at: “In individuals harboring mutations in BRCA1/2 genes, the probability of developing breast cancer over a lifetime is around 85% and that of EOC is about 20–40% [5]”…and following sentences. The authors should expand this paragraph with a discussion on the zygosity of these mutations. Is the 85% breast cancer incidence due to a heterozygous mutation or homozygous? Additionally, the authors should also discuss the various BRCA1/2 mutations. Not all mutations are equal. (see PMID: 35557031).

2.       HR and NHEJ recombination are the two major pathways responsible for repairing DSB, whereas the primary mechanisms for resolving SSB are the BER, NER, MMR and the translesional synthesis [2, 4].” This statement is somewhat simplistic. Single stranded breaks can be converted to DSBs during S-phase. These are repaired by HR particularly BIR. Please expand on this and include relevant literature.

3.       “…whereas HR is error free…” Not all HR is error free. Certain HR sub pathways are not error free (e.g. SSA and even BIR can produce non-reciprocal translocations). The idea that HR is error free is somewhat outdated (maybe before 2010). In fact, now we know that some HR pathways are not error free (PMID: 24966870). Please correct and expand

4.       NHEJ is faster than ΗR and mainly occurs in the G1 phase [9].” This is not exactly true. There is now evidence to suggest that NHEJ functions throughout the cell cycle (see this review PMID: 33785198). In fact, resection factors (e.g MRN) determines the interplay. (see this review PMID: 32648897). Please correct and expand these sections as well.

5.       Single-stranded DNA (ssDNA) generated by resection is coated by replication protein A (RPA), which recruits Ddc2 and the Mec1 checkpoint kinase [12].” This sentence is confusing because you are describing HR in humans in previous paragraphs and here you are using yeast gene nomenclature (e.g. Mec1). Please use ATM and ATR homologues which are humans. Also, the description here is simplistic. It is not quite as simple because before homology search, histones have to be removed. Thus, the next thing that comes after MRN are chromatin remodelers and in fact one chromatin remodeler (KAT5) is activated by ATM. In fact, ATM phosphorylates gamma-H2AX before ssDNA is generated (PMID: 20160506). So how can ATM/Mec1 follow after ssDNA? Again, the point here is that the story is much more complicated and the authors are simplifying it.

6.       Synthetic lethality paragraph. The authors mention important studies but ignore other equally important studies on synthetic lethality. Mutations in the BRCA2-BRCA1-PALB2 axis are lethal with RAD52 (PMID: 21148102, PMID: 33784323, PMID: 26873923, PMID: 33716297, PMID: 30590106, PMID: 22964643). Conversely, certain RAD52 mutations rescue BRCA2 mutations (PMID: 32175645, PMID: 32255263). Please expand this paragraph to include these important findings. To do this, you must have a more through discussion on HR pathways and include discussion of genes such as RAD52, RAD51, RAD54…etc.

7.       Mismatch repair paragraph. This is again simplistic. No mention is made of microsatellite instability (MSI) which actually affects a certain percentage of prostate cancers (PMID: 31974718). Also, please include a discussion on how mismatch repair is often triggered by mutations in POLE.

8.       Beyond BRCA1 paragraph. Here the authors introduce RAD51 paralogs but no effort is made to describe what these paralogs do. Why are there more than one RAD51 gene? Do all the paralogs function in all tissues (you will find that they don’t!!). Also, why don’t we see many germline RAD51 mutations? This is most likely because both BRCA1-BRCA2-PALB2 and RAD52 independently work to load RAD51 onto the resected DNA. Mutations in BRCA2 or RAD52 (independently) are tolerated by mutations in both are lethal. This suggests that RAD51 is absolutely essential for HR and if there is no way to load it, the cells die. Thus, not many cells can survive if they have RAD51 mutations. A great way to explain this is in your synthetic lethality paragraph. Again, this could also be easily explained with a diagram in early paragraphs on how the various recombination genes work.

9.       “PALB2 PVs are quite rare among EOC patients, identified in less than 0.5% of cases interrogated [35, 38].” Why is this so? Because bi-allelic inactivation of PALB2 causes Fanconi anemia which will probably preclude EOC if it ever happens. Mono-allelic inactivation can cause other cancers including EOC (PMID: 30638972). Again, a discussion of zygocity of these mutations as well as BRCA1, BRCA2, RAD52, RAD51, etc.. is absolutely essential in the context of review.

10.   A discussion about chromosomal re-arrangements and cancers should also be included. Do re-arrangements server as drivers of cellular transformation? Are there specific re-arrangements for the cancers discussed here (see PMID: 35091282, PMID: 35804833, PMID: 32024998). This should  be discussed perhaps in the “future directions” section. Some of these re-arrangements do arise because of BRCA or mutations in other repair genes. So is inherited BRCA mutations causing re-arrangements do to inappropriate repair, then the re-arrangements serve as a mutator phenotype that drives immortalization? The role of re-arrangements as potential drivers is certainly becoming obvious and there is a direct connection between BRCA and re-arrangements. Discussing this will increase the relevance of this review.

Minor comments

1.       First line of introduction. What is the evidence that genomic DNA is exposed to a “huge” number of DNA damaging agents? In fact, most DNA damage arises due to endogenous processes, such as DNA replication or transcription. Additional processes such as metabolic byproducts may also cause damage. Exogeneous agents (mutagens and carcinogens) are actually rare particularly in developed countries. Please rephrase sentence because as it is written it sounds like people live in a nuclear fallout zone!! The next two sentences are also awkward. “The damage of DNA increases the mutation rate…” And next sentence: “carcinogenesis is correlated with the impairment of DNA damage repair”. Are you saying that mutations in DNA damage repair genes increases the mutation rate generating a “mutator” phenotype? Please rephrase.

2.       “Among them, BRCA1/2 mutations are the most frequent and hereditary breast and epithelial ovarian cancer (EOC) due to mutations in these genes is the most common cause of hereditary forms of both breast and ovarian cancer, accounting for 30-70% and approximately 90% of cases, respectively [4].” This sentence has two phrases that are redundant.

3.       “HR is restricted to the S and G2 phases of the cell cycle due to the cell cycle-dependent availability of sister chromatids.” The restriction is also due to the fact that HR factors are regulated by the major cyclin dependent kinase (PMID: 35271993).

Author Response

Dear Editor and Reviewers,

I am pleased to resubmit for publication the revised version of cancers-1813589 manuscript, entitled “BRCA Mutations in Ovarian and Prostate Cancer: Bench to Bedside”.

Thankfully the reviewers provided us with a great deal of guidance, regarding how to better position the article. We are hopeful you agree that this revision will update our comprehensive review. All the comments have been addressed, as shown in the revised version of the manuscript, along with this point-by-point response to the reviewers' comments.

All corresponding are blue changes in the manuscript.

Reviewer #2:

  • General comments:

In this review, the authors discuss mutations in BRCA1/2 and related DDR genes and their implications in cancer diagnosis and therapy. The review is relevant and captures some of the literature. However, there are many deficiencies which I point below. Often the authors are too simplistic in their discussion and/or ignore key findings in the field. As a DNA damage repair investigator, I think that this review reads somewhat superficial in this area. I recommend that the authors revise their manuscript in light of my comments below. It is my opinion that such revisions will significantly improve the manuscript and increase its impact.”.

Response:

We appreciate you taking the time to offer us your comments and insights related to the paper. Thank you for your positive reinforcement and constructive feedback. We tried to be responsive to your concerns as we approached our revision.

  • Major comments:

  1. Discussion starting at: “In individuals harboring mutations in BRCA1/2 genes, the probability of developing breast cancer over a lifetime is around 85% and that of EOC is about 20–40% [5]”…and following sentences. The authors should expand this paragraph with a discussion on the zygosity of these mutations. Is the 85% breast cancer incidence due to a heterozygous mutation or homozygous? Additionally, the authors should also discuss the various BRCA1/2 mutations. Not all mutations are equal. (see PMID: 35557031).

Response:

Thank you for your valuable comments, which absolutely makes sense. You meant the “Introduction” section and we have expanded this paragraph appropriately. We have also cited the relevant reference (Reference 7).

  1. HR and NHEJ recombination are the two major pathways responsible for repairing DSB, whereas the primary mechanisms for resolving SSB are the BER, NER, MMR and the translesional synthesis [2, 4].” This statement is somewhat simplistic. Single stranded breaks can be converted to DSBs during S-phase. These are repaired by HR particularly BIR. Please expand on this and include relevant literature.

Response:

Thank you for your recommendation. We have now expanded this paragraph and added 4 references (11-14).

  1. “…whereas HR is error free…” Not all HR is error free. Certain HR sub pathways are not error free (e.g. SSA and even BIR can produce non-reciprocal translocations). The idea that HR is error free is somewhat outdated (maybe before 2010). In fact, now we know that some HR pathways are not error free (PMID: 24966870). Please correct and expand.

Response:

Thank you for your statement. We have now expanded this paragraph and added the relevant reference (16).

  1. NHEJ is faster than ΗR and mainly occurs in the G1 phase [9].” This is not exactly true. There is now evidence to suggest that NHEJ functions throughout the cell cycle (see this review PMID: 33785198). In fact, resection factors (e.g MRN) determines the interplay. (see this review PMID: 32648897). Please correct and expand these sections as well.

Response:

Thank you for the update. We have now expanded this paragraph and added the two recommended references (18, 19).

  1. Single-stranded DNA (ssDNA) generated by resection is coated by replication protein A (RPA), which recruits Ddc2 and the Mec1 checkpoint kinase [12].” This sentence is confusing because you are describing HR in humans in previous paragraphs and here you are using yeast gene nomenclature (e.g. Mec1). Please use ATM and ATR homologues which are humans. Also, the description here is simplistic. It is not quite as simple because before homology search, histones have to be removed. Thus, the next thing that comes after MRN are chromatin remodelers and in fact one chromatin remodeler (KAT5) is activated by ATM. In fact, ATM phosphorylates gamma-H2AX before ssDNA is generated (PMID: 20160506). So how can ATM/Mec1 follow after ssDNA? Again, the point here is that the story is much more complicated and the authors are simplifying it.

Response:

Thank you for your very valuable comments in the terms of these biological processes. We have now incorporated discussion about the ATM and ATR homologues, along with the KAT5 remodeler and the gamma-H2AX. We have also cited two references in this part of the revised manuscript (22 and 25).

  1. Synthetic lethality paragraph. The authors mention important studies but ignore other equally important studies on synthetic lethality. Mutations in the BRCA2-BRCA1-PALB2 axis are lethal with RAD52 (PMID: 21148102, PMID: 33784323, PMID: 26873923, PMID: 33716297, PMID: 30590106, PMID: 22964643). Conversely, certain RAD52 mutations rescue BRCA2 mutations (PMID: 32175645, PMID: 32255263). Please expand this paragraph to include these important findings. To do this, you must have a more through discussion on HR pathways and include discussion of genes such as RAD52, RAD51, RAD54…etc.

Response:

Thank you for your consideration. Undoubtedly, the synthetic lethality is a huge topic and originally we tried to avoid to approach it comprehensively. We have now added the final two paragraphs of that section and cited all the 8 references that you kindly suggested (30-37). The molecular biology information is more thorough now, based on your kind advice.

  1. Mismatch repair paragraph. This is again simplistic. No mention is made of microsatellite instability (MSI) which actually affects a certain percentage of prostate cancers (PMID: 31974718). Also, please include a discussion on how mismatch repair is often triggered by mutations in POLE.

Response:

Thank you for your suggestion. We have now modified the first paragraph and added the second paragraph to discuss the POLE mutations. Two references have been added in total (44 and 45).

  1. Beyond BRCA1 paragraph. Here the authors introduce RAD51 paralogs but no effort is made to describe what these paralogs do. Why are there more than one RAD51 gene? Do all the paralogs function in all tissues (you will find that they don’t!!). Also, why don’t we see many germline RAD51 mutations? This is most likely because both BRCA1-BRCA2-PALB2 and RAD52 independently work to load RAD51 onto the resected DNA. Mutations in BRCA2 or RAD52 (independently) are tolerated by mutations in both are lethal. This suggests that RAD51 is absolutely essential for HR and if there is no way to load it, the cells die. Thus, not many cells can survive if they have RAD51 mutations. A great way to explain this is in your synthetic lethality paragraph. Again, this could also be easily explained with a diagram in early paragraphs on how the various recombination genes work.

Response:

Thank you for your comment, which was very helpful. We have now added the first paragraph and the first two sentences of the second paragraph in the section 3.2, along with figure 3. Moreover, the second and third paragraph of the synthetic lethality section include information associated with RAD51 and RAD52, based on your previous comments.

  1. PALB2 PVs are quite rare among EOC patients, identified in less than 0.5% of cases interrogated [35, 38].” Why is this so? Because bi-allelic inactivation of PALB2 causes Fanconi anemia which will probably preclude EOC if it ever happens. Mono-allelic inactivation can cause other cancers including EOC (PMID: 30638972). Again, a discussion of zygocity of these mutations as well as BRCA1, BRCA2, RAD52, RAD51, etc.. is absolutely essential in the context of review.

Response:

Thank you for your statement. After the sentence that you above mentioned, we added that information, along with the relevant reference (67). The zygocity of these mutations was discussed in the “1. Introduction” and the “3.2. Beyond BRCA1 and BRCA2 genes in ovarian cancer” sections, based on your previous suggestions.

  1. A discussion about chromosomal re-arrangements and cancers should also be included. Do re-arrangements server as drivers of cellular transformation? Are there specific re-arrangements for the cancers discussed here (see PMID: 35091282, PMID: 35804833, PMID: 32024998). This should be discussed perhaps in the “future directions” section. Some of these re-arrangements do arise because of BRCA or mutations in other repair genes. So is inherited BRCA mutations causing re-arrangements do to inappropriate repair, then the re-arrangements serve as a mutator phenotype that drives immortalization? The role of re-arrangements as potential drivers is certainly becoming obvious and there is a direct connection between BRCA and re-arrangements. Discussing this will increase the relevance of this review.

Response:

Thank you for your comment. We have added the first two paragraphs in the “8. Conclusions and future directions” section, along with the relevant references (155-158).

  • Minor comments:

  1. First line of introduction. What is the evidence that genomic DNA is exposed to a “huge” number of DNA damaging agents? In fact, most DNA damage arises due to endogenous processes, such as DNA replication or transcription. Additional processes such as metabolic byproducts may also cause damage. Exogeneous agents (mutagens and carcinogens) are actually rare particularly in developed countries. Please rephrase sentence because as it is written it sounds like people live in a nuclear fallout zone!! The next two sentences are also awkward. “The damage of DNA increases the mutation rate…” And next sentence: “carcinogenesis is correlated with the impairment of DNA damage repair”. Are you saying that mutations in DNA damage repair genes increases the mutation rate generating a “mutator” phenotype? Please rephrase.

Response:

Thank you for your consideration. We have rephrased this part of the introduction as you kindly recommended.

  1. Among them, BRCA1/2 mutations are the most frequent and hereditary breast and epithelial ovarian cancer (EOC) due to mutations in these genes is the most common cause of hereditary forms of both breast and ovarian cancer, accounting for 30-70% and approximately 90% of cases, respectively [4].” This sentence has two phrases that are redundant.

Response:

Thank you. We have now split the sentence and rephrase.

  1. HR is restricted to the S and G2 phases of the cell cycle due to the cell cycle-dependent availability of sister chromatids.” The restriction is also due to the fact that HR factors are regulated by the major cyclin dependent kinase (PMID: 35271993).

Response:

Thank you. We have made the relevant statement and added the recommended reference (15).

Round 2

Reviewer 2 Report

The authors have made significant changes that greatly improved this report. The authors now more comprehensively address DSB repair pathways and expand their analysis to other genes than BRCA2. I have only two minor comments.

In the introduction where the authors talk about endogeneous vs exogeneous sources of damage it would need a reference or two! I suggest this reference: PMID: 33512317. This paper/review shows the rate of DSBs occurring spontaneously. And since the focus is on BRCA2 and associated genes that repair DSBs it would be a good reference. Then the authors can then perhaps state that other signatures, some due to exogenous damage, have also been identified. This website (https://cancer.sanger.ac.uk/signatures/) lists them and gives some references.

In the conclusion where the authors talk about tissue specific mutations/chromosomal re-arrangements due to BRCA gene defficiencies, I suggest referencing these papers: 28256574, 19890832, 30679435,
30089796

Otherwise, in this reviewer's opinion, the authors have done an excellent job revising this paper.

Author Response

Dear Editor and Reviewers,

I am pleased to resubmit for publication the revised version of cancers-1813589 manuscript, entitled “BRCA Mutations in Ovarian and Prostate Cancer: Bench to Bedside”.

The two minor comments of the reviewer “2” have been addressed, as shown in the revised version of the manuscript, along with this point-by-point response.

All corresponding are red changes in the manuscript.

Reviewer #2:

  • General comment:

The authors have made significant changes that greatly improved this report. The authors now more comprehensively address DSB repair pathways and expand their analysis to other genes than BRCA2. I have only two minor comments.”.

Response:

We would like to express our sincere gratitude for your positive reinforcement. Based on your great deal of guidance, the article has been improved. We appreciate the opportunity to revise our work for consideration for publication.

  1. In the introduction where the authors talk about endogeneous vs exogeneous sources of damage it would need a reference or two! I suggest this reference: PMID: 33512317. This paper/review shows the rate of DSBs occurring spontaneously. And since the focus is on BRCA2 and associated genes that repair DSBs it would be a good reference. Then the authors can then perhaps state that other signatures, some due to exogenous damage, have also been identified. This website (https://cancer.sanger.ac.uk/signatures/) lists them and gives some references.

Response:

Thank you for your recommendation. We have added a few sentences associated with the spontaneous DNA damage and the mutational signatures, accompanied by 3 references in total (1-3).

  1. In the conclusion where the authors talk about tissue specific mutations/chromosomal re-arrangements due to BRCA gene defficiencies, I suggest referencing these papers: 28256574, 19890832, 30679435, 30089796.

Response:

Thank you for your suggestion. We have now incorporated in the “8. Conclusions and future directions” section the relevant references (158-161).

Otherwise, in this reviewer's opinion, the authors have done an excellent job revising this paper.

Response:

Thank you; much appreciated.